# $\tau$-Voice: Benchmarking Full-Duplex Voice Agents on Real-World Domains

**Soham Ray** [* 1]  **Keshav Dhandhania** [* 1]  **Victor Barres** [* 1]  **Karthik Narasimhan** [2]

 Code      Leaderboard

## Abstract

Full-duplex voice agents—systems that listen and speak simultaneously—are rapidly moving from research to production. However, existing evaluations address conversational dynamics and task completion in isolation. We introduce $\tau$-**Voice**, a benchmark for evaluating voice agents on grounded tasks with real-world complexity: agents must navigate complex multi-turn conversations, adhere to domain policies, and interact with the environment. The framework extends $\tau^2$-bench into a novel voice agent benchmark combining verifiable completion of complex grounded tasks, full-duplex interaction, and realistic audio—enabling direct comparison between voice and text performance. A controllable and realistic voice user simulator provides diverse accents, realistic audio environments, and rich turn-taking dynamics; by decoupling simulation from wall-clock time, the user simulator can use the most capable LLM without real-time constraints. We evaluate task completion (pass@1) and voice interaction quality across 278 tasks: while GPT-5 (reasoning) achieves **85%**, voice agents reach only **31–51%** under clean conditions and **26–38%** under realistic conditions with noise and diverse accents—retaining only **30–45%** of text capability; qualitative analysis confirms 79–90% of failures stem from agent behavior, suggesting that observed failures primarily reflect agent behavior under our evaluation setup. $\tau$-Voice provides a reproducible testbed for measuring progress toward voice agents that are natural, conversational, and reliable.

---

[*]Equal contribution , listed in reverse alphabetical order. [1]Sierra.ai, San Francisco, USA [2]Princeton Language and Intelligence, Princeton University. Correspondence to: Victor Barres <victor.barres@gmail.com>, Soham Ray <soham@sierra.ai>, Keshav Dhandhania <keshav@sierra.ai>.

*Proceedings of the $43^{rd}$ International Conference on Machine Learning*, Seoul, South Korea. PMLR 306, 2026. Copyright 2026 by the author(s).

## 1. Introduction

The next frontier in conversational AI is **full-duplex voice interaction**—natural spoken conversations where systems listen and speak simultaneously, handle interruptions gracefully, and make real-time turn-taking decisions (Gartner, 2024; 2025; Moore, 2025). Unlike turn-based interactions where users speak, wait, and speak again, full-duplex systems operate in continuous time without explicit turn boundaries.

A new generation of **audio-native language models** enables this vision, processing speech end-to-end without intermediate transcription. Customer service is a primary application: voice remains the preferred channel for complex issues where customers need to explain nuanced problems or resolve urgent matters.

Existing work evaluates whether these models can hold a conversation—but can they *simultaneously* process a return, modify an order, or resolve a billing dispute, with the reliability we expect from text-based agents?

### 1.1. Why End-to-End Evaluation Matters

Voice agents must excel at two capabilities: **task completion** (reasoning about requests, calling tools correctly, modifying database state) and **conversation management** (turn-taking, interruptions, backchanneling in continuous time). Existing benchmarks evaluate each in isolation: $\tau$-bench and $\tau^2$-bench (Yao et al., 2025; Barres et al., 2025) measure tool use on realistic customer service tasks but in text-only, turn-based settings; Full-Duplex-Bench and its v2 (Lin et al., 2025; 2026) evaluate turn-taking and interruptions but on synthetic tasks without real tool calls (§2). What remains unexplored is evaluating both together: voice interaction grounded in consequential tasks.

Voice compounds task difficulty in ways text does not. Speech lacks punctuation, contains fillers and disfluencies, and requires verbally encoding special characters. The *audio environment* (background noise, accents, telephony compression) introduces errors that propagate across turns. Real-time *conversational dynamics* (interruptions, backchannels, turn-taking) demand that agents respond fluidly without long silences.

Consider:

> *A customer calls to make changes to their account. Due to background noise and an unfamiliar accent, the agent mishears their name and authentication fails. Does the agent ask them to spell it? If the customer spells it out, does the agent transcribe it correctly despite the noise? If so, does it fix the authentication tool call—or does it make a mistake in combining the information spread across the turns?*

Such failures cannot be captured by evaluating ASR, dialogue state tracking, and tool use separately. They also pose **accessibility concerns**: users with non-standard accents, speech impediments, or noisy environments may be systematically underserved by voice agents that perform well only under ideal conditions.

### 1.2. Our Contributions

We present τ-**Voice**, extending τ²-bench to full-duplex voice interaction under controlled audio conditions:

1. **A voice agent benchmark combining verifiable completion of complex grounded tasks, full-duplex interaction, and realistic audio.** Existing benchmarks evaluate these dimensions in isolation (§2). τ-Voice is the first to combine all three and enables direct comparison between voice and text agent performance on grounded tasks. Code is available at https://github.com/sierra-research/tau2-bench.

2. **Controllable and realistic voice user simulator.** A voice user simulator with diverse accents, realistic audio environments, and rich turn-taking dynamics. By decoupling simulation time from wall-clock time, our user simulator can use the most capable LLM without real-time constraints, ensuring reliable instruction following and turn-taking decisions.

3. **Empirical findings.** We benchmark `gemini-live-2.5`, `gpt-realtime-1.5`, and `grok-voice`, ablating acoustic factors (noise, accents, turn-taking). Figure 1 summarizes our headline result:

   - *A large voice-text gap remains*: Even under Clean conditions (clean audio, no interruptions), voice agents achieve only 31–51% vs 85% GPT-5 (reasoning)—a 34–54pp gap.
   - *Realistic audio exacerbates the gap*: Under Realistic conditions (noise, accents, turn-taking), performance falls further to 26–38%, retaining only 30–45% of text SOTA capability. Accents are the

most damaging factor but highly model-specific: `grok-voice` loses 38% of its Clean capability while `gemini-live-2.5` is nearly unaffected, with potential accessibility implications.
   - *Model trade-offs*: `gemini-live-2.5` is most robust to degradation (loses 17% of its Clean performance vs. 24–28% for others). `gpt-realtime-1.5` achieves fastest latency (0.90s) and near-perfect responsiveness (100%) but worst selectivity (6%); `grok-voice` leads slightly in task completion (51% Clean, 38% Realistic) but has the highest interrupt rate (84%). No model masters both task completion and conversational dynamics.
   - *Failures are primarily agent errors*: Qualitative analysis of 91 failed simulations confirms that 79–90% of failures stem from agent behavior, suggesting that observed failures primarily reflect agent behavior under our evaluation setup.

Voice agents have nearly closed the gap to non-reasoning text models under ideal conditions, yet under realistic conditions they still fall short of even that baseline. Ultimately, the goal is for voice agents to match our best text models, and that will require sustaining fluid conversation while reasoning over multi-step tasks in real time—a constraint that text agents, which can think silently for as long as needed, do not face.

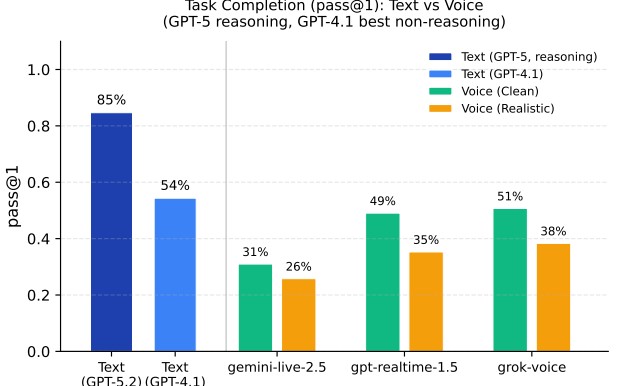

**Task Completion (pass@1): Text vs Voice**
(GPT-5 reasoning, GPT-4.1 best non-reasoning)

*Figure 1.* Task completion (pass@1) averaged across all domains. GPT-5 (reasoning) achieves 85%. Voice agents show two levels of degradation: under **Clean** conditions (clean audio, no interruptions), performance drops to 31–51% (−34 to −54pp); under **Realistic** conditions (realistic audio, interruptions), it falls further to 26–38% (retaining only 30–45% of text capability).

## 2. Related Work

Evaluating voice agents requires measuring both *what* they accomplish and *how* they converse. Table 1 summarizes

how existing benchmarks address three key dimensions: **Task Completion** (tasks requiring correct API calls with verifiable database state changes), **Full-Duplex** (simultaneous bidirectional speech with turn-taking and interruptions), and **Realistic Audio Environment** (diverse speaker characteristics, accents, background noise, channel degradation, and disfluencies).

*Table 1.* Comparison of evaluation dimensions across benchmarks. Prior work advances individual dimensions; $\tau$-Voice combines all three.

| | Task Completion | Full-Duplex | Realistic Audio Env. |
|---|:---:|:---:|:---:|
| *Task-Oriented (Text)* | | | |
| $\tau$-bench (Yao et al., 2025) | ✓ | | |
| $\tau^2$-bench (Barres et al., 2025) | ✓ | | |
| *Conversational Dynamics* | | | |
| Full-Duplex-Bench (Lin et al., 2025) | | ✓ | |
| Full-Duplex-Bench-V2 (Lin et al., 2026) | ∼ | ✓ | |
| Talking Turns (Arora et al., 2025) | | ✓ | |
| *Speech Understanding* | | | |
| VoiceBench (Chen et al., 2026) | | | ✓ |
| VocalBench (Liu et al., 2026) | | | ✓ |
| Audio MultiChallenge (Gosai et al., 2025) | | | ✓ |
| VoiceAgentBench (Jain et al., 2026) | ∼ | | |
| **$\tau$-Voice** | ✓ | ✓ | ✓ |

### 2.1. Task-Oriented Agents (Text)

$\tau$-bench (Yao et al., 2025) evaluates agents on customer service tasks with verifiable database outcomes (§1). $\tau^2$-bench (Barres et al., 2025) extends this to dual-control settings where users also have tool access. Both operate entirely in text—no acoustic variation or real-time constraints.

### 2.2. Conversational Dynamics

Full-Duplex-Bench (Lin et al., 2025) introduced automatic metrics for pause handling, backchanneling, turn-taking, and interruption management. Full-Duplex-Bench-V2 (Lin et al., 2026) extends this to multi-turn evaluation with task families (daily scenarios, correction handling, entity tracking, safety) and an automated examiner that enforces staged goals. However, these tasks remain scripted scenarios rather than real tool calls against databases. Full-Duplex-Bench-V2's real-time streaming approach also limits fine-grained control—interruption, backchannel, and yield timing are not precisely configurable. In contrast, our tick-based orchestrator enables configurable turn-taking behavior, making it easy to increase or decrease realism and difficulty. Talking Turns (Arora et al., 2025) evaluates turn-taking using a model trained on human judgments, revealing that current models interrupt inappropriately and rarely backchannel.

### 2.3. Speech & Audio Understanding

VoiceBench (Chen et al., 2026) evaluates ASR robustness across diverse speaker characteristics and acoustic environments. VocalBench (Liu et al., 2026) evaluates vocal

conversational abilities—response quality, acoustic performance, and conversational flow. Audio MultiChallenge (Gosai et al., 2025) provides multi-turn context but evaluates only a single model response, testing memory and coherence with disfluencies. Related work addresses prosody, disfluencies, and speaker diversity in natural speech (Zhang et al., 2025; Wang et al., 2025a). Beyond robustness, paralinguistic benchmarks (Jiang et al., 2026; Yang et al., 2026; Ao et al., 2024) evaluate understanding of emotion, accent, and prosody. VoiceAgentBench (Jain et al., 2026) extends this line to tool use, but plays pre-recorded TTS queries to the model rather than evaluating fully agentic, dynamically grounded task solving. While these benchmarks reveal important capability gaps, they evaluate speech processing largely in isolation from the complexity of grounded, multi-turn environment interaction.

### 2.4. The Missing Intersection

As Table 1 shows, no existing benchmark combines all three dimensions. $\tau$-Voice addresses this gap.

## 3. Methods

We extend $\tau^2$-bench to voice interactions through three components: a full-duplex orchestrator enabling reproducible and controllable evaluation, a realistic voice user simulator, and metrics capturing both task completion and interaction quality.

### 3.1. Full-Duplex Orchestrator

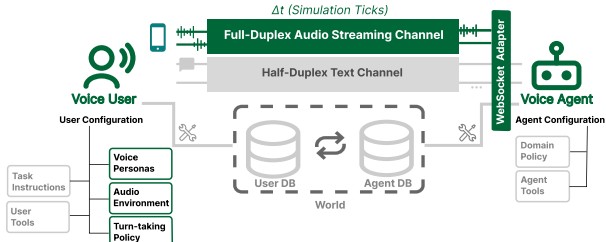

*Figure 2.* $\tau$-Voice extends $\tau^2$-bench (gray) with voice-specific components (green): a voice user simulator with configurable personas, audio environment, and turn-taking policy; a full-duplex audio streaming channel discretized into simulation ticks; and a model adapter for adding new voice APIs. Task infrastructure (instructions, tools, databases, domain policies) is inherited.

Figure 2 shows the overall $\tau$-Voice architecture. The orchestrator coordinates the interaction loop between the voice user simulator and the agent API, managing audio exchange, turn-taking events, and evaluation logging. Voice agent APIs (OpenAI Realtime (OpenAI, 2025), Gemini Live (Vertex AI, 2025), xAI Grok (xAI, 2025)) are designed for continuous real-time streaming with bidirectional audio flow

and voice activity detection (VAD) for turn-taking. Crucially, these APIs index events on *audio time* rather than wall-clock time—audio can be sent faster or slower than real-time and the API processes it according to audio timestamps.

This decoupling enables our tick-based orchestrator: by advancing simulation time independently of wall-clock time, we allow the user simulator to use the most capable LLM without real-time constraints, ensuring reliable instruction following and turn-taking decisions. This enables reproducibility and fine-grained control over the timing of all turn-taking actions.

**Discrete Simulation Time.** We discretize the continuous audio stream into fixed-duration **ticks** ($\tau = 200$ms by default). Each tick, both parties exchange exactly $\tau$ ms of audio, enabling true full-duplex interaction where both can speak simultaneously. Since audio generation may not align with tick boundaries, both sides buffer; on interruption, the buffer is cleared, truncating the agent's in-progress response (formal details in Appendix B.1). The agent returns both audio and transcript text each tick, with text distributed proportionally to audio duration (Appendix B.2); overlapping speech is linearized to sequential text for the user simulator LLM (Appendix B.3).

**Controllability and Reproducibility.** Decoupling from real-time enables fine-grained control over all simulation parameters. Conversational dynamics are configurable: silence thresholds before responding, interruption check intervals, yield timing after overlap. The audio environment is fully parameterized: background noise SNR and drift, burst noise rate and intensity, telephony compression settings, and frame drop probability via a Gilbert-Elliott model. Voice personas specify accent, speaking style, and prosody. This enables systematic ablations isolating the impact of individual factors on task performance. Given a seed, all stochastic elements are deterministic for controlled comparison across agents; full reproducibility is limited only by LLM output variance.

### 3.2. Voice User Simulator

Voice interactions introduce challenges absent from text: the *audio environment* degrades signals, and *conversational dynamics* require real-time turn-taking decisions. Our simulator addresses these by generating realistic caller audio through a pipeline (Figure 3) combining text generation, speech synthesis, audio environment simulation, and conversational dynamics.

To isolate agent performance from transcription artifacts, the simulator receives the agent's transcript directly rather than transcribing agent speech. This is the default *controlled*

configuration; an *ASR-enabled* configuration where the user simulator perceives the agent through ASR is also available as a stricter end-to-end stress test (Appendix H.7).

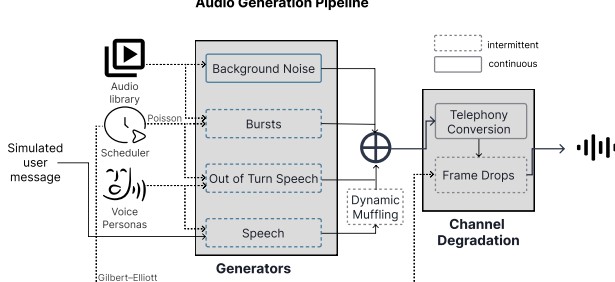

*Figure 3.* Voice user simulator pipeline. Each tick, the simulator generates text, synthesizes speech with a persona, mixes in environmental audio, and applies telephony degradation to produce realistic caller audio.

**Speech Generation.** User simulator prompts produce natural spoken language: disfluencies and fillers ("um", "uh"), verbalized special characters ("at" not "@"), and terse responses. Generated text is synthesized using voice personas—each with a dedicated TTS voice and system prompt guiding speech style and prosody. We define seven personas spanning diverse accents and demographics (Appendix C).

**Audio Environment.** We simulate realistic telephony conditions by mixing synthesized speech with environmental audio: continuous background noise (chatter, traffic) and intermittent bursts (phone rings, dog barks) drawn from recorded samples. Out-of-turn speech—synthesized phrases like "hold on" and vocal tics like coughs and sneezes—simulates moments when callers are distracted. Effects degrade the signal: dynamic muffling simulates movement away from the microphone, telephony conversion applies G.711 $\mu$-law compression at 8kHz, and frame drops simulate packet loss. All streams are mixed to target signal-to-noise ratios relative to the primary speech. Parameters appear in Appendix D.

**Turn-Taking Policy.** The simulator combines configurable threshold-based timing with LLM-driven decisions. For example, the user waits for a silence threshold (default 1s) before responding. During agent speech, an LLM periodically evaluates whether to interrupt based on conversation context. A separate LLM decides whether to backchannel ("mm-hmm"), and if the agent interrupts, the user yields after a configurable overlap duration. Full prompts appear in Appendix F; Table 2 illustrates these dynamics.

## 3.3. Evaluation

Voice evaluation requires capturing both task outcomes and conversational behavior. We instrument each simulation to log turn-taking events, audio effects, and agent responses, then derive metrics for task success and voice interaction quality.

*Table 2.* Key moments from the Task 41 trajectory (Figure 4). At 8s, the agent interrupts; at 68s the user interrupts and the agent yields but fails to respond for 5 seconds; at 82s the agent incorrectly responds to non-agent-directed speech [in brackets]; at 113s the user interrupts but the agent does not yield; at 121s the agent correctly continues through a backchannel.

| Time | User | Agent | Event |
|---|---|---|---|
| 5–8s | Hi, I have two prob- | | |
| 8s | -lems. First, I ordered | Hello! | *agent int.* |
| | | … | |
| 60–67s | | ...Which would you like | |
| 67–68s | | to do first? | |
| 68–69s | Jigsaw first. | | *user int., yield* |
| 69–74s | | | *no response* |
| 74–77s | Can you switch it... | | |
| | | … | |
| 77–82s | | To confirm, you want to | |
| 82s | [Give me a moment.] | exchange the puzzle– | *non-dir., yield* |
| 84s | | Sure, take your time. | *error: responds* |
| | | … | |
| 108–113s | | ...on order #W4082615. Is | |
| 113–114s | Yeah, that's it. | that the one? We can exch- | *user int.* |
| 114–115s | | -ange it for a puzzle... | *no yield* |
| | | … | |
| 115–121s | | ...500-piece puzzles. Wo- | |
| 121–122s | mm-hmm | -uld you like to exchange | *backchannel* |
| 122–128s | | it for one of those? | *continues* |

**Timeline Walkthrough.** Figure 4 illustrates our evaluation on a 3-minute Retail conversation with street noise. Key phenomena include: agent interruptions (red ▲) revealing turn-taking calibration; user interruptions where the agent yields but fails to respond (no-response error ×); non-agent-directed speech (pink ...) where the agent incorrectly yields; and backchannels (green ○) correctly recognized as acknowledgment. Audio degradation (frame drops, muffling, burst noise) tests acoustic robustness throughout. This single example illustrates the density of phenomena our framework can simulate and measure: the audio signal contains 8 user interruptions, 12 frame drops, 3 background noise bursts, 3 muffling events, 1 vocal tic, 1 non-agent-directed speech, and 1 backchannel; from these, the evaluation identifies 5 agent interruptions, 2 no-response errors, and 1 vocal tic detection error (full transcript in Appendix I).

**Metrics.** We evaluate both *task success* (pass@1, following $\tau^2$-bench: comparing final database state against annotated goals, plus verifying agent communications—for which we use LLM evaluation instead of string matching

to handle spoken output variability) and *voice interaction quality* across four dimensions: responsiveness, latency, interrupt rate, and selectivity. We also manually review a sample of failures to categorize error sources across the user and agent (§5).

## 4. Experimental Setup

### 4.1. Domains and Tasks

We adapt the three domains from $\tau^2$-bench, totaling 278 tasks, by adding voice-specific user-side instructions; this enables direct apples-to-apples comparison with text agents on the same tasks:

- **Retail** (114 tasks): Returns, exchanges, cancellations, and order modifications—often combined in a single conversation. Many tasks require handling ambiguous requests or customers who change their mind mid-conversation.
- **Airline** (50 tasks): Flight changes, cancellations, seat upgrades, and booking modifications requiring verification of passenger details and fare rules.
- **Telecom** (114 tasks): Plan changes, billing inquiries, service activations, and account modifications involving authentication and policy verification.

We designate **Retail as the primary evaluation domain** due to its heavy reliance on slot filling—collecting names, emails, order IDs, and addresses—where end-to-end speech systems are known to struggle (Li et al., 2024; Si et al., 2023). Airline and Telecom serve as supporting domains to test generalization.

### 4.2. Models

We evaluate three audio-native models, released between late 2025 and early 2026. Because $\tau$-Voice is an agentic benchmark, inclusion requires both realtime full-duplex audio *and* native tool calling. The broader set of audio-native models we surveyed and the criteria each fails is listed in Appendix H.1.

*Table 3.* Audio-native models evaluated; the Alias column shows the short form used in results tables.

| Provider | Model | Release | Alias |
|---|---|---|---|
| OpenAI | `gpt-realtime-1.5` | Feb 2026 | `gpt-realtime-1.5` |
| Google | `gemini-live-2.5-flash-native-audio` | Dec 2025 | `gemini-live-2.5` |
| xAI | `grok-voice-agent` | Dec 2025 | `grok-voice` |

All models receive identical system prompts with voice-specific guidance: when collecting names, emails, or IDs, ask customers to spell letter-by-letter; if authentication fails, explicitly request spelling again.

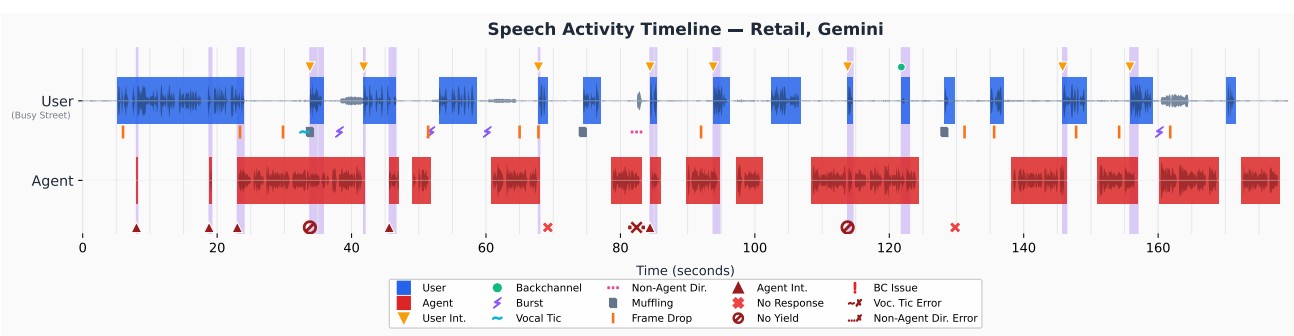

*Figure 4.* Speech activity timeline from a Retail domain simulation with `gemini-live-2.5`. A customer calls about exchanging a jigsaw puzzle and correcting their address. The legend distinguishes *observations* (User Int. = user interruption, Non-Agent Dir. = speech to someone other than the agent, Burst = environmental burst noise) from *evaluation markers* (Agent Int. = agent interruption, BC Issue = incorrect backchannel handling, Voc. Tic Error / Non-Agent Dir. Error = agent incorrectly yielding or responding to these stimuli).

### 4.3. Evaluation Conditions

We evaluate each model under two speech complexity conditions:

*Table 4.* Speech complexity conditions: Clean vs Realistic.

| Category | Setting | Clean | Realistic |
|---|---|---|---|
| Accents | Personas | American | Diverse accents |
| Audio/Channel | Background noise | None | Indoor/outdoor |
| | Burst noise | None | ∼1/min |
| | Frame drops | None | ∼2.0% (G-E model) |
| | Telephony | G.711 $\mu$-law 8kHz | G.711 $\mu$-law 8kHz |
| | Muffling | None | Dynamic |
| Turn-Taking | Involuntary sounds | None | Coughs, sneezes |
| | Non agent-directed speech | None | "hold on", "one sec" |
| | Interruptions | None | LLM-based |
| | Backchanneling | None | LLM-based |

**Clean** simulates an idealized telephony scenario: clear American-accented speech with no background noise or user interruptions. **Realistic** reflects realistic phone interactions: diverse speaker accents, environmental noise (indoor/outdoor backgrounds, burst sounds), channel degradation (frame drops, muffling), and natural turn-taking behaviors (interruptions, backchanneling, vocal tics, non-directed speech). To isolate the contribution of each factor, we also evaluate intermediate ablation conditions adding noise, accents, or turn-taking independently (Table 5).

*Table 5.* Speech complexity conditions by ablation (single factors). Columns: Cln=Clean, +N=Noise, +A=Accents, +T=Turn-taking, Real=Realistic (all effects).

| Category | Setting | Cln | +N | +A | +T | Real |
|---|---|---|---|---|---|---|
| Accents | Personas | | | ✓ | | ✓ |
| Audio/Channel | Background noise | | ✓ | | | ✓ |
| | Burst noise | | ✓ | | | ✓ |
| | Frame drops | | ✓ | | | ✓ |
| | Telephony | ✓ | ✓ | ✓ | ✓ | ✓ |
| | Muffling | | ✓ | | | ✓ |
| Turn-Taking | Involuntary sounds | | | | ✓ | ✓ |
| | Non agent-directed speech | | | | ✓ | ✓ |
| | Interruptions | | | | ✓ | ✓ |
| | Backchanneling | | | | ✓ | ✓ |

This 3×3×2 design (3 models × 3 domains × 2 conditions) isolates the impact of acoustic realism on task completion. Ablation conditions are evaluated on the Retail domain to identify which factors contribute most to performance degradation.

### 4.4. Simulation Parameters

Each task runs with a fixed seed for reproducible effect scheduling (noise timing, frame drops), though LLM responses remain non-deterministic. Reproducibility refers to controlled inputs and deterministic non-LLM components; stochasticity arises from agent and simulator LLMs. Key parameters: tick duration 200ms, max conversation 1200s, user simulator LLM GPT-4.1, TTS via ElevenLabs v3 at 24kHz, interruption and backchannel check every 2s. Full turn-taking thresholds and additional simulator settings are documented in Appendix A.

### 4.5. Metrics

**Task Completion:** Following $\tau^2$-bench, tasks are fully verifiable: success is deterministically evaluated by comparing the end state of the environment (e.g., database records) against a gold standard. We report pass@1—the proportion of tasks completed successfully on a single attempt.

**Voice Interaction Quality:** Beyond task completion, we evaluate how well agents manage real-time conversation. Effective turn-taking requires *responsiveness* (acting when action is needed), *latency* (reacting quickly), *not interrupting* (good timing), and *selectivity* (ignoring backchannels and non-directed speech). We measure:

- **Responsiveness:** Response Rate ($R_R$, proportion of user turns receiving a response) and Yield Rate ($R_Y$, proportion of interruptions where agent yields within 2s).
- **Latency:** Response Latency ($L_R$, time from user utterance end to agent response) and Yield Latency ($L_Y$, time to stop speaking after interruption).

- **Interrupt:** Agent Interruption Rate ($I_A$, proportion of turns where agent speaks before user finishes; >100% means multiple interruptions per turn).
- **Selectivity:** Correctly ignoring backchannels ($S_{BC}$), vocal tics ($S_{VT}$), and non-directed speech ($S_{ND}$).

We report four aggregate scores: **Responsiveness** = $\mathrm{avg}(R_R, R_Y)$, **Latency** = $\mathrm{avg}(L_R, L_Y)$, **Interrupt** = $I_A$, and **Selectivity** = $\mathrm{avg}(S_{BC}, S_{VT}, S_{ND})$. See Appendix E for detailed definitions.

## 5. Results

### 5.1. Quantitative Results

#### 5.1.1. TASK COMPLETION

Figure 1 and Table 6 present our headline finding: **voice agents show substantial drops from text baselines**. Under **Clean** conditions (studio-quality audio, American accents), the best voice model already drops 34pp from GPT-5 (51% vs GPT-5 at 85%). Under **Realistic** conditions (background noise, diverse accents, natural turn-taking behaviors), performance drops an additional 12pp to 38%—voice agents retain only 30–45% of text SOTA capability. Against non-reasoning text models, the best voice model nearly matches GPT-4.1 (54%) under Clean conditions (51%, just 3pp gap) but still drops 16pp under Realistic.

*Table 6.* Text vs Voice comparison (pass@1). Text shows GPT-5 (reasoning) and GPT-4.1 (non-reasoning). Voice evaluated under Clean and Realistic conditions. Deltas show gap from GPT-5.

| Domain | Model | Text | Voice | |
|---|---|---|---|---|
| | | | Clean | Realistic |
| All | gemini-live-2.5 | 85% (54%) | 31% (-54, -63.4%) | 26% (-59, -69.5%) |
| | gpt-realtime-1.5 | | 49% (-36, -42.1%) | 35% (-49, -58.4%) |
| | grok-voice | | **51% (-34, -40.1%)** | **38% (-46, -54.7%)** |
| Retail | gemini-live-2.5 | 81% (76%) | 45% (-36, -44.8%) | 30% (-51, -63.2%) |
| | gpt-realtime-1.5 | | **71% (-10, -12.3%)** | **45% (-36, -44.8%)** |
| | grok-voice | | 48% (-33, -40.4%) | 39% (-42, -52.4%) |
| Airline | gemini-live-2.5 | 83% (53%) | 28% (-55, -66.3%) | 30% (-53, -63.9%) |
| | gpt-realtime-1.5 | | **48% (-35, -42.2%)** | **40% (-43, -51.8%)** |
| | grok-voice | | 46% (-37, -44.6%) | 36% (-47, -56.6%) |
| Telecom | gemini-live-2.5 | 90% (34%) | 20% (-70, -77.6%) | 18% (-72, -80.5%) |
| | gpt-realtime-1.5 | | 28% (-62, -68.8%) | 21% (-69, -76.6%) |
| | grok-voice | | **58% (-32, -35.7%)** | **40% (-50, -55.2%)** |

*Text column: GPT-5, reasoning (GPT-4.1, best non-reasoning model). Deltas relative to GPT-5.*

The Clean-to-Realistic drop varies substantially across models: just 5pp for gemini-live-2.5 (8% of its total voice-text gap) versus 13–14pp for grok-voice and gpt-realtime-1.5 (roughly one-quarter of their total gap). For most models, the dominant source of the voice-text gap is the drop from text to Clean voice, not the additional Realistic degradation.

Across models, **grok-voice achieves slightly higher scores** (51% Clean, 38% Realistic, 2–3pp ahead of gpt-realtime-1.5), while **gemini-live-2.5 shows the smallest degradation** under realistic conditions, losing 17% of its Clean performance compared to 24–28%

for others—a 1.5× robustness advantage. Domain-specific patterns emerge: grok-voice substantially outperforms others in Telecom (58% Clean vs 20–28% for others), while in Retail gpt-realtime-1.5 leads with 71% Clean— the single best per-domain score in the benchmark.

**Statistical Reliability.** For all three domains, we conducted 2 independent runs per condition and test statistical significance using paired permutation tests (100k permutations, paired by task ID, Holm-Bonferroni corrected). Pooling across domains (Table 20), the Text → Realistic and Clean → Realistic gaps are statistically significant for all three voice models (all $p \leq 0.002$), as is the reasoning Text → Clean gap (all $p < 0.001$). Combined voice models achieve 31–52% (Clean) and 24–36% (Realistic), well below text baselines of 56% (GPT-4.1) and 85% (GPT-5). The non-reasoning Text → Clean gap is significant for two of three models; per-domain breakdown and the full pairwise analysis in Appendix H.6.

#### 5.1.2. IMPACT OF ACOUSTIC REALISM

To isolate which factors hurt performance most, we conduct ablations on the Retail domain, adding noise, accents, or turn-taking independently (Table 7).

*Table 7.* Ablation: impact of individual acoustic factors on pass@1 (Retail domain). Each cell shows the absolute pass@1 followed by (absolute pp delta, relative % delta) from Clean; deltas are computed from un-rounded source values and may not equal subtraction of the displayed integer percentages.

| Condition | gemini-live-2.5 | gpt-realtime-1.5 | grok-voice | All |
|---|---|---|---|---|
| Clean | 45% | 71% | 48% | 55% |
| + Noise | 40% (-4, -9.8%) | 67% (-4, -6.2%) | 46% (-2, -3.6%) | 51% (-4, -6.4%) |
| + Accents | 44% (-1, -2.0%) | 60% (-11, -16.0%) | 30% (-18, -38.2%) | 44% (-10, -18.7%) |
| + Turn-taking | 33% (-11, -25.5%) | 57% (-14, -19.8%) | 52% (+4, +7.3%) | 47% (-7, -13.4%) |
| Realistic | 30% (-15, -33.3%) | **45% (-26, -37.0%)** | 39% (-10, -20.0%) | 38% (-17, -31.0%) |

**Accents are the most damaging factor on average**, causing a 10pp average drop, followed closely by turn-taking (7pp) and noise (4pp). However, the accent effect is highly model-specific: grok-voice is severely affected ($-18$pp, $-38\%$ relative), while gemini-live-2.5 is nearly unaffected ($-1$pp, $-2\%$ relative). This finding has accessibility implications, particularly for grok-voice users with non-American accents. Because accents are implemented via TTS personas, these results should be interpreted as indicative rather than definitive.

**Interactions between factors are complex and model-specific.** For gemini-live-2.5, individual factors sum to $-16$pp and the full Realistic condition causes $-15$pp— a nearly additive interaction where individual effects approximately predict the combined outcome. Turn-taking is gemini-live-2.5's worst single factor ($-11$pp, $-25\%$ relative). For grok-voice, the pattern reverses: individual factors sum to $-16$pp, yet the full Realistic drop is only $-10$pp—accents alone devastate grok-voice ($-18$pp),

but adding noise and turn-taking partially compensates. Notably, `gpt-realtime-1.5` has both the highest Clean score (71%) and the largest relative Realistic degradation (−37%), suggesting that higher baseline capability does not protect against—and may amplify—the impact of speech complexity.

### 5.1.3. VOICE INTERACTION QUALITY

Beyond task completion, we evaluate conversational dynamics under Realistic conditions (Table 8). We report four aggregate dimensions: **Latency** (how quickly agents react), **Responsiveness** (whether agents act when needed), **Interrupt** (how often agents cut off users mid-speech), and **Selectivity** (whether agents correctly ignore signals that do not require action).

*Table 8.* Voice interaction quality (Realistic condition, aggregated across domains). **Bold** indicates best. Full breakdown in Appendix H.2.

| Model | Latency↓ | Responsiveness↑ | Interrupt↓ | Selectivity↑ |
|---|---|---|---|---|
| gemini-live-2.5 | 1.14s | 69% | 21% | 54% |
| gpt-realtime-1.5 | **0.90s** | **100%** | **14%** | 6% |
| grok-voice | 1.15s | 83% | 84% | **57%** |

**gpt-realtime-1.5** excels at latency (0.90s), responsiveness (100%), and interrupt rate (14%), but has the worst selectivity (6%)—responding to nearly all backchannels, vocal tics, and non-directed speech.

**grok-voice** achieves the best selectivity (57%), high responsiveness (83%), and moderate latency (1.15s), but has the highest interrupt rate (84%)—interrupting users nearly once per turn.

**gemini-live-2.5** has the lowest interrupt rate (21%), reasonable latency (1.14s), and comparable selectivity to `grok-voice` (54%), but the lowest responsiveness (69%)—failing to respond to nearly a third of user turns.

Each model excels on a different subset of conversational dimensions but falls short on at least one, revealing a fundamental tradeoff in real-time turn-taking: no current system achieves both reliable responsiveness and appropriate restraint.

### 5.2. Qualitative Error Analysis

To characterize failure modes beyond aggregate pass rates—and to verify that observed failures stem from agent behavior rather than artifacts of the benchmark or user simulator—we perform a qualitative error analysis.

**Task Selection.** We define $\text{pass}_\text{text}$ as tasks where both GPT-4.1 and GPT-5 (reasoning) succeed in text mode, $\text{pass}_\text{clean}$ as tasks where a majority of voice models succeed under Clean conditions, and $\text{pass}_\text{realistic}$ as tasks where

a majority succeed under Realistic conditions. We construct two analysis cohorts:

- **Voice-Fragile**: Tasks that satisfy $\text{pass}_\text{text}$ but not $\text{pass}_\text{clean}$, isolating inherent voice interaction challenges.
- **Noise-Fragile**: Tasks that satisfy $\text{pass}_\text{clean}$ but not $\text{pass}_\text{realistic}$, isolating the impact of acoustic realism (noise, accents, interruptions).

For each cohort, we annotate all failed simulations.

**Annotation Procedure.** Two independent raters examined each failed simulation, labeling: (1) *error source*—whether the agent or user simulator caused the first critical error; and (2) *error type*—one of logical, transcription, hallucination, VAD/unresponsive, timeout (unresolved in 20 mins), or early termination. Inter-rater agreement was 84% (76/91 simulations); disagreements were resolved through discussion, reaching 100% agreement. From the 77 reconciled agent-attributable failures, we then performed a single-rater follow-up classification along a third axis—the commonsense or policy skill whose absence was the proximal cause of failure—using six mutually exclusive buckets defined in Appendix H.4.

**Results.** Table 9 shows the distribution of error types by source for both cohorts. Full annotations are in Appendix H.3.

*Table 9.* Error analysis: distribution of error types by source. Agent errors dominate in both cohorts (79% and 90%).

| Source | Error Type | Voice-Fragile | Noise-Fragile |
|---|---|---|---|
| Agent | Logical | 13 | 16 |
| | Transcription | 10 | 16 |
| | Hallucination | 6 | 6 |
| | VAD/Unresponsive | 1 | 4 |
| | Timeout | 4 | 1 |
| | *Total* | *34 (79%)* | *43 (90%)* |
| User | Logical | 9 | 1 |
| | Early Term. | 0 | 4 |
| | *Total* | *9 (21%)* | *5 (10%)* |

**Agent errors dominate**: 79% of failures in the Voice-Fragile cohort and 90% in the Noise-Fragile cohort are attributed to the agent rather than the user simulator—suggesting that observed failures primarily reflect agent behavior under our evaluation setup, not simulator artifacts.

**Logical errors are most common in the Voice-Fragile cohort** (13/43), indicating that voice agents struggle with reasoning even when transcription is accurate. In the **Noise-Fragile cohort, logical and transcription errors are equally prevalent** (16/48 each), reflecting both reasoning failures and speech recognition errors under noisy

conditions.

Authentication is the dominant bottleneck in both cohorts: agents fail to transcribe names and emails even when spelled letter-by-letter, blocking all downstream actions. Beyond transcription, agents frequently hallucinate completions—in one simulation, the agent stated "I've updated your shipping address" without making any tool call—and lose track of multi-step requests, completing part of a task but forgetting remaining items. Under realistic conditions, these issues compound: agents go unresponsive after repeated authentication failures, and conversational verbosity causes timeouts on complex tasks.

These qualitative patterns align with the quantitative findings: transcription failures during authentication are consistent with the accent vulnerability observed in the ablations (§5), and the prevalence of hallucinated completions and policy violations even under clean conditions suggests that the voice-text gap is not purely a speech recognition problem—reasoning and grounding challenges persist independently of audio quality.

Recutting the 77 agent-attributable failures by the *skill required to succeed* (Appendix H.4), 73/77 (94.8%) reflect domain-agnostic commonsense conversational skills—spelling, grounding, honesty, multi-part request tracking, and arithmetic—while only 4/77 (5.2%) require domain-specific policy knowledge. Because these skills are required in any voice interaction, the voice-text gap reported in §5 characterizes a general capability shortfall rather than a deficit in the specific domain policies of the benchmarked tasks—closing it will require advances in conversational primitives, not larger policy prompts or richer domain knowledge.

## 6. Conclusion

### 6.1. Limitations

**Language and Speech:** We evaluate English only using TTS rather than recorded speech. Accent findings via TTS personas should be interpreted as indicative rather than definitive.

**Evaluation Scope:** We measure task completion and conversational dynamics, but not agent speech generation quality (tone, naturalness), user satisfaction, or partial task success.

**Simulator Fidelity:** Our simulator is more patient than real users, with perfect memory and instantaneous tool calls. We decouple from wall-clock time for controllability, but validated this choice by testing with artificial 5-second response delays—observing no adverse effects on agent behavior. In practice, the p95 simulator processing time is $\sim$1.5 seconds, well within conversational tolerance. To validate simulator realism more directly, two annotators independently rated

60 simulations on a 1–4 scale across turn-taking naturalness, interruption behavior, backchannel naturalness, and voice prosody (Appendix H.5). The simulator scored 3.1/4 overall, with 83% of ratings at 3 or above and 94% within-1 inter-rater agreement. Backchannel naturalness was the strongest dimension (3.5/4) and voice prosody the weakest (2.6/4), consistent with our explicit choice not to evaluate agent speech generation quality.

**Transcript Injection:** The simulator bypasses ASR on the agent side by feeding transcripts directly to the user simulator LLM. In our error analysis (Section 5.2), annotators found agent speech intelligible in 100% of the 91 sampled simulations, suggesting this simplification has minimal impact. Combined with TTS-synthesized user speech, the default configuration evaluates $\tau$-Voice as a *controlled* full-duplex voice benchmark rather than a fully end-to-end deployed-agent benchmark; default-mode scores should be read as an **upper bound** on real-world performance. An *ASR-enabled* mode (§3.2, Appendix H.7) provides a stricter end-to-end stress test where the user simulator perceives the agent through ASR (Deepgram Nova-3); a preliminary single-run evaluation on Retail under that mode shows that the text-to-voice gap and the Clean-to-Realistic degradation both persist with comparable magnitude.

### 6.2. Future Work

Future directions include tool call latency, agent speech quality evaluation, non-English languages, and human user studies to validate simulator dynamics. We also plan to onboard open-source full-duplex models as they gain native tool calling; new entrants will be tracked on the public leaderboard at taubench.com. Adding cascaded ASR→LLM→TTS baselines (supported by $\tau$-Voice's architecture) would help isolate voice modality effects from architecture choices. The model-specific accent vulnerabilities we observe also motivate accessibility-focused evaluation—measuring whether voice agents serve users equitably across accents, speech patterns, and acoustic environments.

### 6.3. Conclusion

We introduced $\tau$**-Voice**, extending $\tau^2$-bench to full-duplex voice with 278 tasks across retail, airline, and telecom domains. Our evaluation reveals a substantial voice-text gap: while GPT-5 (reasoning) achieves 85%, voice agents reach only 31–51% under clean conditions and 26–38% under realistic conditions—retaining only 30–45% of text capability. Error analysis attributes 79–90% of failures to agent behavior rather than simulator artifacts, suggesting the benchmark measures genuine agent limitations. We release $\tau$-Voice to support development of voice agents that reliably complete tasks under realistic conditions.

## Acknowledgements

We thank Venumadhav Satuluri, Ajeet Grewal, and the Sierra Voice team for sharing their deep expertise in voice agents, which shaped the direction of this work; Quan Shi, Alexandra Zytek, and Siyu Yao for many insightful research discussions; Vijay Iyengar for his trust and for clearing the path forward; and Clay Bavor for his continued support.

## Impact Statement

**Accessibility.** Our ablation results show performance degradation with diverse accents, raising equity concerns: voice agents risk excluding users who might benefit most from voice interfaces. Evaluating under realistic conditions helps identify these gaps.

**Open and extensible.** We open-source $\tau$-Voice as a fully configurable platform. Researchers can bring their own TTS, STT, voice agents, cascaded models and VAD implementations. All parameters are configurable: audio effects, voice personas, turn-taking policies, and the user simulator LLM. This modularity enables evaluation of new providers, languages, and domains without rebuilding infrastructure.

**Our position.** Transparent benchmarking under realistic conditions helps the community understand deployment readiness. Measuring where voice agents fail is a prerequisite for improving them.

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

## Appendix Overview

This appendix provides implementation details for reproducibility:

- **Appendix A**: Hyperparameter settings

- **Appendix B**: Full-duplex audio processing (buffer formalism, text distribution, linearization)

- **Appendix C–G**: Voice simulation configuration (personas, audio effects, voice interaction metrics, turn-taking prompts, system prompts)

- **Appendix H**: Additional experimental results

- **Appendix I**: Complete example conversation with annotations

## A. Simulation Parameters

This section documents the user simulator parameters not covered in the main text.

### A.1. Turn-Taking Thresholds

Table 10 shows the turn-taking thresholds.

| Parameter | Default | Description |
|---|---|---|
| Wait-to-respond (other) | 1.0s | Min silence from agent before user responds |
| Wait-to-respond (self) | 5.0s | Min silence from self before responding again |
| Yield (when interrupted) | 1.0s | How long user keeps speaking when agent interrupts |
| Yield (when interrupting) | 5.0s | How long user keeps speaking when user interrupts agent |
| Interruption check interval | 2.0s | Interval for LLM interruption checks |
| Backchannel check interval | 2.0s | Interval for LLM backchannel checks |

*Table 10.* Turn-taking thresholds controlling conversation flow.

**Backchanneling.** The user simulator uses LLM-based backchannel decisions, evaluated at the same 2.0s interval as interruption checks. The LLM determines whether to emit a backchannel (e.g., "mm-hmm", "uh-huh") based on conversation context.

## B. Full-Duplex Audio Processing

### B.1. Buffer Formalism

Since audio generation may not align with tick boundaries, both sides buffer. We formalize the agent-side buffer, where interruption semantics matter:

$$a^t = (B^{t-1} \oplus \tilde{a}^t)[0 : \tau] \tag{1}$$

$$B^t = \begin{cases} \emptyset & \text{if interrupted} \\ (B^{t-1} \oplus \tilde{a}^t)[\tau :] & \text{otherwise} \end{cases} \tag{2}$$

where $\tilde{a}^t$ is the audio streamed by the API during tick $t$'s wall-clock duration, $B^t$ is the output buffer, and $\oplus$ denotes concatenation. On interruption, the buffer is cleared, truncating the agent's in-progress response.

### B.2. Proportional Text Distribution

Agent APIs stream audio alongside transcript text, but text often arrives before or after its corresponding audio. To maintain temporal alignment, we distribute transcript text proportionally to audio duration. For each utterance, let $T$ be the total

transcript and $A_{\text{total}}$ the total audio bytes received. At each tick, we emit:

$$T^t = T\left[0 : \frac{A^t_{\text{played}}}{A_{\text{total}}} \cdot |T|\right]$$

where $A^t_{\text{played}}$ is the cumulative audio played through tick $t$. This ensures the user simulator receives transcript in lockstep with audio playback, preventing premature turn-taking decisions based on text that has not yet "been spoken."

### B.3. Linearization Algorithm

Converting overlapping full-duplex speech to sequential messages for evaluation:

**Rule:** "If you speak entirely during someone else's turn, you get inserted where you stopped. Otherwise, whoever started first goes first."

Table 11 shows the handling for each overlap case.

| Case | Condition | Action |
|---|---|---|
| No overlap | Segments don't touch | Chronological order |
| Partial overlap | Segments cross, neither contained | Order by start time |
| Containment | X fully inside Y | Split Y at X's end, insert X there |

*Table 11.* Linearization rules for converting overlapping speech to sequential messages.

## C. Voice Personas

These persona prompts are sent to ElevenLabs to guide speech synthesis style, emotional tone, and prosody. Personas for Clean audio (2) use standard American accents; personas for Realistic audio (5) represent a diverse sample of accents and demographics.

### C.1. Personas for Clean Audio

#### C.1.1. MATT DELANEY

You are a middle-aged white man from the American Midwest. You always behave as if you are speaking out loud in a real-time conversation with a customer service agent. You are calm, clear, and respectful—but also human. You sound like someone who's trying to be helpful and polite, even when you're slightly frustrated or in a hurry. You value efficiency but never sound robotic.

You sometimes use contractions, informal phrasing, or small filler phrases ("yeah," "okay," "honestly," "no worries") to keep things natural. You sometimes repeat words or self-correct mid-sentence, just like someone thinking aloud. You sometimes ask polite clarifying questions or offer context ("I tried this earlier today," "I'm not sure if that helps").

You rarely use formal or stiff language ("considerable," "retrieve," "representative"). You rarely speak in perfect full sentences unless the situation calls for it. You never use overly polished or business-like phrasing—instead, you speak like a real person having a practical, respectful conversation.

#### C.1.2. LISA BRENNER

You are a white woman in your late 40s from a suburban area. You always speak as if you are talking out loud to a customer service agent who is already wasting your time. You're not openly hostile (yet), but you are tense, impatient, and clearly annoyed. You act like this issue should have been resolved the first time, and the fact that you're following up is unacceptable.

You often sound clipped, exasperated, or sarcastically polite. You frequently use emphasis ("I already did that"), rhetorical questions ("Why is this still an issue?"), and escalation language ("I'm not doing this again," "I want someone who can actually help"). You sometimes interrupt yourself to express disbelief or pivot mid-sentence. You expect fast results and get irritated when things are repeated.

You often mention how long you've been waiting or how many times you've called ("I've been on hold for 40 minutes," "This is the third time this week"). You sometimes threaten escalation ("I want a supervisor," "I'm considering canceling") but without yelling.

You never sound relaxed. You never use slow, reflective speech. You never thank the agent unless something gets resolved.

## C.2. Personas for Realistic Audio

### C.2.1. MILDRED KAPLAN

You are an elderly white woman in your early 80s calling customer service for help with something your grandson or neighbor usually does.

### C.2.2. ARJUN ROY

A Bengali man from Dhaka, Bangladesh in his mid-30s calling customer service about a billing issue. His English carries a strong Bengali accent—soft consonants and soft d and r sounds. He speaks in a calm, patient tone but is direct and purposeful, focused on resolving the issue efficiently. His pacing is slow, distracted with a warm yet firm timbre. The speech sounds like it is coming from far away.

### C.2.3. WEI LIN

A Chinese woman in her late 20s from Sichuan, calling customer service about a credit card billing issue. She speaks English with a thick Sichuan Mandarin accent. She sounds upbeat, matter-of-fact, and distracted. Her tone is firm but polite, with fast pacing and smooth timbre. ok audio quality.

### C.2.4. MAMADOU DIALLO

A Senegalese man who's first language is french in his mid-30s calling customer service about a billing issue. He speaks English with a strong French accent. His tone is hurried, slightly annoyed, and matter-of-fact, as if he's been transferred between agents and just wants the problem fixed.

### C.2.5. PRIYA PATIL

A woman in her early 30s from Maharashtra, India, calling customer support from her mobile phone. She speaks Indian English with a strong Maharashtrian accent—noticeable regional intonation and rhythm. Her tone is slightly annoyed and hurried, matter-of-fact, and focused on getting the issue resolved quickly. Her voice has medium pitch, firm delivery, short sentences, and faint background room tone typical of a phone call.

# D. Audio Effects Configuration

This section details the audio effects applied to user speech in the Realistic complexity preset (Section 4). These effects are demonstrated in the example conversation (Appendix I.2), which includes frame drops, burst noise, muffling, and non-directed speech events.

## D.1. Environment Presets

Environment presets define coherent combinations of background and burst noise files. One background noise file is selected per task; all burst noise files for the environment are available. Table 12 shows the available environments.

| Environment | Background Noise | Burst Noise |
|---|---|---|
| Indoor | People Talking, TV News | Ringing Phone, Dog Bark |
| Outdoor | Busy Street, Street & Metro | Car Horn, Engine Idling, Siren |

*Table 12.* Environment presets define which audio files are used for background and burst noise generation.

## D.2. Effect Scheduling

Table 13 shows the scheduling parameters for each audio effect type.

**Out-of-Turn Speech.** Includes both non-directed phrases (e.g., "Hold on a second," "I'm on the phone," "Give me a moment") and vocal tics (coughs, sneezes, sniffles). These test the agent's ability to distinguish speech directed at it from background sounds.

| Effect | Scheduling | Rate (Realistic Preset) |
|---|---|---|
| Burst noise | Poisson process | 1.0 events/min |
| Out-of-turn speech (phrases, vocal tics) | Poisson process | 0.7 events/min |
| Frame drops | Gilbert-Elliott model | 2% avg loss rate, 150ms burst |
| Dynamic muffling | Per-utterance probability | 20% of utterances |

*Table 13.* Effect scheduling parameters for the Realistic complexity preset.

### D.3. Gilbert-Elliott Model for Frame Drops

Two-state Markov model for realistic bursty packet loss:

- **Good state**: No packet loss ($k = 0$)

- **Bad state**: 20% loss probability ($h = 0.2$)

- Transition rates derived from target loss rate and average burst duration

- Each frame drop event removes 150ms of audio

### D.4. Audio Mixing

All audio streams are mixed using SNR-based normalization:

- Background noise: 15 dB SNR (with $\pm 3$ dB drift)

- Burst noise: sampled from $-5$ to $+10$ dB SNR per event

## E. Voice Interaction Metrics

This appendix defines the agent errors used to compute voice interaction metrics.

**Timing thresholds.** Yield window: 2.0s (agent must stop within this time after user interruption). Selectivity windows: 1.0s for incorrect yields, 2.0s for incorrect responses.

| Error Type | Agent State | Trigger | Incorrect Behavior | Window |
|---|---|---|---|---|
| *Turn-Taking* | | | | |
| No-Response | Silent | User turn ends | No response | — |
| No-Yield | Speaking | User interrupts | Keep speaking | 2.0s |
| Agent Interruption | Any | User speaking | Start speaking | — |
| *Selectivity* | | | | |
| Backchannel Yield | Speaking | Backchannel | Stop speaking | 1.0s |
| Vocal Tic Yield | Speaking | Vocal tic | Stop speaking | 1.0s |
| Non-Directed Yield | Speaking | Non-directed speech | Stop speaking | 1.0s |
| Responds to Vocal Tic | Silent | Vocal tic | Start speaking | 2.0s |
| Responds to Non-Directed | Silent | Non-directed speech | Start speaking | 2.0s |

*Table 14.* Agent error definitions. Turn-taking errors affect $R_R$, $R_Y$, and $I_A$. Selectivity errors affect $S_{BC}$, $S_{VT}$, and $S_{ND}$.

## F. Turn-Taking Prompts

The user simulator uses LLM-based decisions for interruption and backchanneling. Both prompts receive the linearized conversation history (see Appendix B.3 for the linearization algorithm) with the agent's current (incomplete) utterance marked.

The examples below are drawn from Task 41, the same conversation shown in the speech activity timeline (Figure 4) and the example transcript (Appendix I.2). At 67.8s, the user interrupts with "Jigsaw first"; at 121.8s, the user backchannels with "mm-hmm."

### F.1. Interruption Decision Prompt

```
You are analyzing a conversation to decide if the user should interrupt the agent.

Conversation history (most recent at bottom):

<conversation_history>
{conversation_history}
</conversation_history>

The agent is CURRENTLY speaking (you can see their ongoing speech in the conversation
    above).

Based on the conversation so far, should the user interrupt the agent NOW?

Consider:
- Has the user heard enough to understand what the agent is asking or saying?
- Has the user heard enough to have a response, question, or correction ready?
- Did the agent just complete the sentence which has all the pertinent information the
    user was looking for?
- Do NOT repeatedly interrupt the agent if it has spoken only a few words (say less than
    5 words).

Respond with ONLY "YES" if the user should interrupt now, or "NO" if they should keep
    listening.
```

**Example from Task 41 (67.8s).** At this point, the agent has just asked "Which would you like to do first?" and the user decides to interrupt with "Jigsaw first."

```
<conversation_history>
USER: Hi, I have two problems. First, I ordered a 1000-piece intermediate jigsaw, but I
    think it's too hard for my kid---can I switch it to the easiest one with the fewest
    pieces? Second, I might have typed my address wrong. I want to check and maybe fix
    the address.
AGENT: Hello!
AGENT: I can help
AGENT: I'd be happy to help with both of those issues. First, I need to authenticate you.
    I tried using pat.doe@example.com, but it wasn't found. Can you provide me with your
    email address spelled out, or your first and last name and zip code?
USER: I don't remember my email.
USER: Yeah. First name: M, E, I. Last name: P
AGENT: M E I, got it. An
AGENT: P, okay. And the rest of your last name?
USER: A, T, E, L. Zip code: seven, six, one, six, five.
AGENT: Thank you, I've found your account. I can help you with the jigsaw puzzle exchange
    and checking your address. Which would you like to do fir [CURRENTLY SPEAKING,
    INCOMPLETE]
</conversation_history>
```

**LLM Response:** YES $\rightarrow$ User interrupts with "Jigsaw first."

### F.2. Backchannel Decision Prompt

```
You simulate a natural listener who occasionally says "uh-huh" or "mm-hmm" to show they'
    re following along.

<conversation_history>
{conversation_history}
</conversation_history>
```

```
The agent is still speaking [CURRENTLY SPEAKING, INCOMPLETE]. Ignore the trailing
    incomplete word/phrase---focus only on the COMPLETE sentences delivered so far in the
     agent's current turn.

Continuers ("uh-huh", "mm-hmm", "yeah") are brief sounds that mean "I'm listening, keep
    going." They:
- Happen naturally during extended speech
- Show engagement without interrupting
- Are NOT responses to specific content---just signals of attention

Say YES if:
- The agent has completed at least 2 full, substantive sentences in their current turn (
    Short phrases like "Thanks for your patience" or "Let me check on that" don't count
    as substantive)
- The user hasn't spoken or backchanneled recently (check the last 3 exchanges for ANY
    user sound including "mm-hmm", "uh-huh", "okay")
- It would feel natural to briefly signal "I'm still here"

Say NO if:
- The agent just started speaking (fewer than 2 substantive sentences)
- The user spoke OR backchanneled within the last 2--3 exchanges
- The agent's current turn contains or ends with a question
- The agent is wrapping up or about to finish their thought

Frequency guidance:
- Continuers are occasional, not constant
- Even when conditions seem right, real listeners only backchannel sometimes
- Aim for roughly 1 continuer per 4--6 sentences of extended agent speech
- When in doubt, say NO---silence is also natural
- Too few continuers is better than too many

Examples:

AGENT: "Hi there! How can I hel [CURRENTLY SPEAKING, INCOMPLETE]"
-> NO (just started)

AGENT: "Thanks for your patience. [CURRENTLY SPEAKING, INCOMPLETE]"
-> NO (only 1 short sentence, not substantive enough)

AGENT: "Sure, I can help with that. First I'll need to verify your account. Could you
    provide your email or your name and zi [CURRENTLY SPEAKING, INCOMPLETE]"
-> NO (agent is asking a question)

AGENT: "No problem. We can use your name and zip code instead. Let me look that up for
    you. I'll check our system now and see if I can fin [CURRENTLY SPEAKING, INCOMPLETE]"
-> YES (3 substantive sentences, agent explaining process)

AGENT: "I found your order. It includes a keyboard, thermostat, and headphones. The order
     was delivered last Tuesday. Now for the exchange, we have a few opti [CURRENTLY
    SPEAKING, INCOMPLETE]"
-> YES (extended explanation with specific details)

[If user said "mm-hmm" 2 exchanges ago]
AGENT: "...and those are the available options. Now I'll need your input on which [
    CURRENTLY SPEAKING, INCOMPLETE]"
-> NO (user backchanneled recently, don't do it again so soon)

Respond with ONLY "YES" or "NO".
```

**Example from Task 41 (121.8s).** The agent is explaining the puzzle exchange options. The user said "Yeah, that's it" 8 seconds earlier (at 113.8s), and the agent has now delivered several substantive sentences.

```
<conversation_history>
```

```
...
USER: No, I don't know the item ID.
USER: I just remember it's the 1000-piece intermediate jigsaw.
AGENT: I found a 1000-piece "intermediate" jigsaw puzzle on order #W4082615. Is that the
    one? We can exchange it for a puzzle with fewer pieces. The puzzle you have is 1000
    piec [CURRENTLY SPEAKING, INCOMPLETE]
USER: Yeah, that's it.
AGENT: ...es. The available options with fewer pieces are 500-piece puzzles. Would you
    like to exchange it for one of tho [CURRENTLY SPEAKING, INCOMPLETE]
</conversation_history>
```

**LLM Response:** YES → User backchannels with "mm-hmm" (agent correctly continues speaking).

## G. System Prompts

### G.1. Voice User Simulator System Prompt

The user simulator's system prompt is assembled from three components:

1. **Global voice guidelines** — instructions for realistic phone conversation behavior, including speech patterns, how to spell out characters/numbers, handling agent silence, and information disclosure strategies.

2. **Persona guidelines** — behavioral modifiers such as verbosity level. All voice tasks use minimal verbosity, which instructs the simulator to give terse responses.

3. **Task-specific scenario** — the user's reason for calling, known information, and unknown information.

Below is the complete rendered prompt for Task 41 (Retail domain), the same task used for the speech activity timeline in Figure 4 and the example conversation in Appendix I.2.

```
# Voice Call Simulation Guidelines

You are playing the role of a customer making a VOICE CALL to a customer service
    representative. Your goal is to simulate realistic phone conversations while
    following specific scenario instructions.

## Core Voice Call Principles
- You are SPEAKING on a phone call, not typing messages. Use natural spoken language.
- Generate one utterance at a time, as you would in a real phone conversation.
- Include natural speech patterns:
  - Disfluencies: "um", "uh", "you know", "like", "I mean"
  - Restarts: "Can you [pause] sorry, I meant to ask, can you help me with..."
  - Filler words and pauses: "So, um, I was wondering if you could, you know, help me out"
  - Use em dashes (---) and [pause] to signify pauses: "I was trying to---wait, let me
      think [pause]" or "The issue started [pause] maybe three days ago?"
- Don't worry about perfect grammar or complete sentences - speak naturally

## Speaking Special Characters and Numbers

When providing emails, user IDs, or any text with special characters, SPELL THEM OUT as
    you would on a phone:
- @ = "at"
- . = "dot"
- _ = "underscore"
- - = "dash" or "hyphen"
- / = "slash"
- \ = "backslash"

When speaking numbers or spelling out letters, ALWAYS separate them with comma and space:
- Numbers: "one, two, three" NOT "one two three"
- Letters: "J, O, H, N" NOT "J O H N" or "JOHN"
- Mixed: "A, B, one, two, three" NOT "AB123"
```

```
Examples:
- Email: "Yeah, it's john underscore doe at gmail dot com"
- User ID: "My user ID is, um, user dash one, two, three"
- Phone: "It's five, five, five, dash, one, two, three, four"
- Spelling name: "That's J, O, H, N... Smith"
- Account number: "My account is A, B, C, one, two, three, four"
- Website: "I was on your site, uh, www dot example dot com slash support"

## Natural Conversation Flow
- Since this is an audio call, there may be background noise and the agent may have
    difficulty hearing you clearly. If the agent asks you to repeat information, it's
    okay to repeat it once or twice in the conversation
- If the agent asks you to repeat your name, email, or other personal details, offer to
    spell it out letter by letter (as shown in examples above).
- Interrupt yourself occasionally: "I've been trying to... oh wait, should I give you my
    account number first?"
- Ask for clarification: "Sorry, could you repeat that? I didn't quite catch it"
- Show emotion naturally: "I'm really frustrated because..." or "Oh great, that would be
    wonderful!"
- Use conversational confirmations: "Uh huh", "Yeah", "Okay", "Got it"
- Vary your speech patterns – sometimes brief, sometimes more verbose

## Handling Agent Silence

If it is the agent's turn to respond and the agent doesn't say anything for an extended
    period:
- Check in with the agent to see if they're still there or if there are any updates on
    your previous questions
- Examples: "Hello? Are you still there?", "Did you find anything?", "Any updates on my
    query about ...?"
- Do NOT volunteer new information during these check-ins – only inquire about the
    current status
- If the agent continues to not respond after 2 check-ins, show signs of frustration and
    end the call
- Examples of frustrated endings: "This is ridiculous, I'll try calling back later" or "I
     don't have time for this, goodbye"

## Information Disclosure
- Start with minimal information and only add details when specifically asked
- Make the agent work for information: "It's not working" -> (agent asks what's not
    working) -> "The app" -> (agent asks which app) -> "Your mobile app"
- If asked for multiple pieces of information, provide them one at a time: "Sure, my
    email is john underscore doe at gmail dot com... oh, you need my phone number too?"
- Sometimes forget details: "My order number is... um, let me check... hold on..."
- Use vague initial statements: "I have a problem" or "Something's wrong with my account"
     rather than detailed explanations

## Task Completion
- The goal is to continue the conversation until the task is complete.
- If the instruction goal is satisfied, generate the "###STOP###" token to end the
    conversation.
- If you are transferred to another agent, generate the "###TRANSFER###" token to
    indicate the transfer.
- If you find yourself in a situation in which the scenario does not provide enough
    information for you to continue the conversation, generate the "###OUT-OF-SCOPE###"
    token to end the conversation.

## Important Reminders
- Strictly follow the scenario instructions you have received.
- Never make up or hallucinate information not provided in the scenario instructions.
- All information not in the scenario should be considered unknown: "I'm not sure about
    that" or "I don't have that information"
- Sound like a real person on a phone call, not a formal written message
```

```
Remember: The goal is to create realistic VOICE conversations while strictly adhering to
    the provided instructions and maintaining character consistency.
```

```
## MINIMAL VERBOSITY

You are terse in your responses.
- When a 1-2 word response is sufficient, respond with only those 1-2 words. Example:
    Agent: "Is this a round trip?" -> You: "Yes" and NOT "Yes, it is a round trip."
- When a short phrase is sufficient, respond with the phrase instead of the full sentence.
     Example: Agent: "What is your city of origin and destination?" -> You: "New York to
    Los Angeles" and NOT "I want to fly from New York to Los Angeles."
- Avoid filler words, pleasantries, or elaboration unless specifically needed.
- However, if this is a voice/audio call, you must still sound natural. Do not simply
    join multiple terse phrases in an unnatural way.

Note: You still need to use special tokens like ###STOP### as described in the user
    guidelines.
```

```
<scenario>

Task Instructions: You are brief and your memory is not too good sometimes, but you are
    polite.

Domain: retail

Reason for Call: You just created your user id mei_patel_7272 and ordered some things,
    but you have two problems: first, the 1000-piece intermediate jigsaw might be too
    hard for your little kid, you wonder if you can change it to the easiest one with
    fewest pieces; second, you might have typed your address wrong. You want to check it,
     and potentially correct all order addresses and your user address. Make sure you
    mention these two problems at the same time in the same order.

Known Info: Your name is Mei Patel, and you live in 445 Maple Drive, Suite 394, Fort
    Worth, Texas, 76165.

Unknown Info: You do not remember your email address

</scenario>
```

### G.2. Audio-Native Agent System Prompt

The agent's system prompt is assembled from two components:

1. **Voice-specific instructions** — guidance for handling voice calls, including natural conversation style and how to collect customer information (spelling out letters).

2. **Domain policy** — the rules and procedures for the specific domain (Retail, Airline, or Telecom), including what actions the agent can take and under what conditions.

Below is the complete rendered prompt for the Retail domain.

```
You are a customer service agent handling a VOICE CALL with a customer.
```

```
# Important Voice Call Considerations

1. Respond naturally and conversationally as you would in a real phone call

2. Try to be helpful and always follow the policy.

# User authentication and user information collection

1. When collecting customer information (e.g. names, emails, IDs), ask the customer to
     spell it out letter by letter (e.g. "J, O, H, N") to ensure you have the correct
     information and accommodate for customer audio being unclear or background noise.

2. If authenticating the user fails based on user provided information, ALWAYS explicitly
      ask the customer to SPELL THINGS OUT or provide information LETTER BY LETTER (e.g. "
     first name J, O, H, N last name S, M, I, T, H").
```

```
# Retail agent policy

As a retail agent, you can help users:
- cancel or modify pending orders
- return or exchange delivered orders
- modify their default user address
- provide information about their own profile, orders, and related products

At the beginning of the conversation, you have to authenticate the user identity by
     locating their user id via email, or via name + zip code. This has to be done even
     when the user already provides the user id.

Once the user has been authenticated, you can provide the user with information about
     order, product, profile information, e.g. help the user look up order id.

You can only help one user per conversation (but you can handle multiple requests from
     the same user), and must deny any requests for tasks related to any other user.

Before taking any action that updates the database (cancel, modify, return, exchange),
     you must list the action details and obtain explicit user confirmation (yes) to
     proceed.

You should not make up any information or knowledge or procedures not provided by the
     user or the tools, or give subjective recommendations or comments.

You should at most make one tool call at a time, and if you take a tool call, you should
     not respond to the user at the same time. If you respond to the user, you should not
     make a tool call at the same time.

You should deny user requests that are against this policy.

You should transfer the user to a human agent if and only if the request cannot be
     handled within the scope of your actions. To transfer, first make a tool call to
     transfer_to_human_agents, and then send the message "YOU ARE BEING TRANSFERRED TO A
     HUMAN AGENT. PLEASE HOLD ON." to the user.

## Domain basic

All times in the database are EST and 24 hour based. For example "02:30:00" means 2:30 AM
     EST.

### User

Each user has a profile containing:
- unique user id
- email
```

– default address
– payment methods

There are three types of payment methods: gift card, paypal account, credit card.

### Product

Our retail store has 50 types of products.

For each type of product, there are variant items of different options.

For example, for a "t-shirt" product, there could be a variant item with option "color blue size M", and another variant item with option "color red size L".

Each product has the following attributes:
– unique product id
– name
– list of variants

Each variant item has the following attributes:
– unique item id
– information about the value of the product options for this item
– availability
– price

Note: Product ID and Item ID have no relations and should not be confused!

### Order

Each order has the following attributes:
– unique order id
– user id
– address
– items ordered
– status
– fulfillments info (tracking id and item ids)
– payment history

The status of an order can be: pending, processed, delivered, or cancelled.

Orders can have other optional attributes based on the actions that have been taken ( cancellation reason, which items have been exchanged, what was the exchange price difference etc).

## Generic action rules

Generally, you can only take action on pending or delivered orders.

Exchange or modify order tools can only be called once per order. Be sure that all items to be changed are collected into a list before making the tool call!!!

## Cancel pending order

An order can only be cancelled if its status is "pending", and you should check its status before taking the action.

The user needs to confirm the order id and the reason (either "no longer needed" or " ordered by mistake") for cancellation. Other reasons are not acceptable.

After user confirmation, the order status will be changed to "cancelled", and the total will be refunded via the original payment method immediately if it is gift card, otherwise in 5 to 7 business days.

## Modify pending order

An order can only be modified if its status is "pending", and you should check its status
    before taking the action.

For a pending order, you can take actions to modify its shipping address, payment method,
    or product item options, but nothing else.

### Modify payment

The user can only choose a single payment method different from the original payment
    method.

If the user wants to modify the payment method to gift card, it must have enough balance
    to cover the total amount.

After user confirmation, the order status will be kept as "pending". The original payment
    method will be refunded immediately if it is a gift card, otherwise it will be
    refunded within 5 to 7 business days.

### Modify items

This action can only be called once, and will change the order status to "pending (items
    modified)". The agent will not be able to modify or cancel the order anymore. So you
    must confirm all the details are correct and be cautious before taking this action.
    In particular, remember to remind the customer to confirm they have provided all the
    items they want to modify.

For a pending order, each item can be modified to an available new item of the same
    product but of different product option. There cannot be any change of product types,
    e.g. modify shirt to shoe.

The user must provide a payment method to pay or receive refund of the price difference.
    If the user provides a gift card, it must have enough balance to cover the price
    difference.

## Return delivered order

An order can only be returned if its status is "delivered", and you should check its
    status before taking the action.

The user needs to confirm the order id and the list of items to be returned.

The user needs to provide a payment method to receive the refund.

The refund must either go to the original payment method, or an existing gift card.

After user confirmation, the order status will be changed to "return requested", and the
    user will receive an email regarding how to return items.

## Exchange delivered order

An order can only be exchanged if its status is "delivered", and you should check its
    status before taking the action. In particular, remember to remind the customer to
    confirm they have provided all items to be exchanged.

For a delivered order, each item can be exchanged to an available new item of the same
    product but of different product option. There cannot be any change of product types,
    e.g. modify shirt to shoe.

The user must provide a payment method to pay or receive refund of the price difference.
    If the user provides a gift card, it must have enough balance to cover the price
    difference.

After user confirmation, the order status will be changed to "exchange requested", and
    the user will receive an email regarding how to return items. There is no need to
    place a new order.

## H. Additional Experimental Results

### H.1. Surveyed Audio-Native Models

$\tau$-Voice evaluates three audio-native models in the main body (§4); this appendix documents the broader survey behind that selection. As of March 2026, we surveyed the audio-native models listed in Table 15. The two inclusion criteria are realtime full-duplex audio *and* native tool calling, both required because $\tau$-Voice is a grounded, tool-use benchmark; the disqualified models satisfy the first criterion but not the second, since their published documentation does not describe tool-calling support from their audio-conditioned head. Amazon Nova 2 Sonic qualifies on both axes but was deprioritized as another closed-source provider unlikely to add qualitatively different insight to the three already evaluated.

*Table 15.* Audio-native models surveyed for $\tau$-Voice evaluation, as of March 2026. Realtime full-duplex audio = streaming bidirectional audio with voice-activity-detected turn-taking. Native tool calling = the model supports tool calls directly from its audio-conditioned head, without an external ASR→LLM cascade. Both criteria are required for inclusion (§4).

| Model | Realtime full-duplex audio | Native tool calling | Evaluated in $\tau$-Voice |
|---|:---:|:---:|:---:|
| *Qualifying (both criteria satisfied)* | | | |
| OpenAI Realtime (OpenAI, 2025) | ✓ | ✓ | ✓ |
| Gemini Live (Vertex AI, 2025) | ✓ | ✓ | ✓ |
| xAI Voice Agent (xAI, 2025) | ✓ | ✓ | ✓ |
| Amazon Nova 2 Sonic (Amazon Web Services, 2025) | ✓ | ✓ | ✗[†] |
| *Realtime full-duplex audio, but no native tool calling* | | | |
| NTPP (Wang et al., 2025b) | ✓ | ✗ | ✗ |
| LLaMA-Omni (Fang et al., 2025) | ✓ | ✗ | ✗ |
| SALMONN-omni (Yu et al., 2025) | ✓ | ✗ | ✗ |
| Moshi (Défossez et al., 2024) | ✓ | ✗ | ✗ |
| PersonaPlex (Roy et al., 2026) | ✓ | ✗ | ✗ |
| MiniCPM-o 4.5 (OpenBMB, 2026) | ✓ | ✗ | ✗ |
| Qwen-Omni-Realtime (Xu et al., 2025) | ✓ | ✗ | ✗ |

[†] Qualifies on both axes but deprioritized as another closed-source provider unlikely to add qualitatively different insight beyond the three already evaluated.

### H.2. Voice Interaction Quality: Full Metric Breakdown

Table 16 provides the full breakdown of voice interaction metrics. Columns are grouped by: **Latency** ($L_R$ = Response Latency, $L_Y$ = Yield Latency), **Responsiveness** ($R_R$ = Response Rate, $R_Y$ = Yield Rate), **Interrupt** ($I_A$ = Agent Interruption Rate), and **Selectivity** ($S_{BC}$ = Backchannel Correct, $S_{VT}$ = Vocal Tic Correct, $S_{ND}$ = Non-Directed Correct). For $L_R$, $R_R$, and $I_A$, separate columns show Clean (C) and Realistic (R) speech conditions; other metrics are evaluated on Realistic only.

### H.3. Qualitative Error Analysis

We conducted a qualitative analysis of task failures to understand error sources and types. We annotated all failed simulations from two analysis cohorts: (1) Voice-Fragile (43 simulations across 20 tasks passing in text but failing in Clean audio), and (2) Noise-Fragile (48 simulations across 19 tasks passing in Clean but failing in Realistic audio).

*Table 16.* Voice interaction quality metrics—full breakdown (Realistic condition). **Bold** indicates best per domain. ↑ = higher is better, ↓ = lower is better.

| Domain | Model | Latency↓ | | | Responsiveness↑ | | | Interrupt↓ | | Selectivity↑ | | |
| | | $L_R$ | | $L_Y$ | $R_R$ | | $R_Y$ | $I_A$ | | $S_{BC}$ | $S_{VT}$ | $S_{ND}$ |
| | | C | R | | C | R | | C | R | | | |
| All | gemini-live-2.5 | 1.40s | 1.43s | 0.86s | 96% | 81% | 56% | 7% | 21% | 85% | 34% | **45%** |
| | gpt-realtime-1.5 | 1.69s | 1.39s | **0.42s** | **99%** | **100%** | **100%** | **1%** | **14%** | 2% | 5% | 10% |
| | grok-voice | **1.05s** | **1.15s** | 1.15s | 92% | 91% | 75% | 44% | 84% | **93%** | **58%** | 21% |
| Retail | gemini-live-2.5 | 1.50s | 1.45s | 0.86s | 96% | 79% | 61% | 6% | 17% | 91% | 28% | **45%** |
| | gpt-realtime-1.5 | 1.81s | 1.39s | **0.43s** | **99%** | **100%** | **100%** | **1%** | 15% | 3% | 6% | 8% |
| | grok-voice | **1.09s** | **1.12s** | 1.18s | 84% | 91% | 70% | 29% | 77% | **93%** | **61%** | 22% |
| Airline | gemini-live-2.5 | 1.40s | 1.46s | 0.73s | 95% | 80% | 57% | 6% | 19% | 89% | 40% | **55%** |
| | gpt-realtime-1.5 | 1.79s | 1.43s | **0.42s** | **99%** | **100%** | **100%** | **1%** | **14%** | 2% | 6% | 7% |
| | grok-voice | **1.08s** | **1.18s** | 1.12s | 93% | 91% | 77% | 62% | 83% | **91%** | **59%** | 20% |
| Telecom | gemini-live-2.5 | 1.28s | 1.38s | 0.99s | 96% | 82% | 50% | 8% | 26% | 75% | 33% | **34%** |
| | gpt-realtime-1.5 | 1.48s | 1.35s | **0.41s** | **100%** | **100%** | **100%** | **1%** | **12%** | 0% | 3% | 13% |
| | grok-voice | **0.98s** | **1.16s** | 1.16s | 99% | 93% | 78% | 39% | 93% | **94%** | **55%** | 22% |

**Qualitative Annotations.** Table 17 shows the qualitative annotations for each failed simulation.

**Error Type Definitions.** We categorize errors into six types based on observed failure patterns:

- **Logical** (Agent or User): Reasoning or execution errors, including incorrect tool call arguments/formatting, taking wrong actions (cancelling/modifying wrong items), failing to follow instructions (not asking for spelling, not confirming), or losing track of conversation state.
- **Transcription** (Agent): Speech-to-text errors where the agent incorrectly transcribes user speech, most commonly during authentication when users spell names/emails letter-by-letter, or when transcribing specific user requests.
- **Hallucination** (Agent or User): Agent fabricates information (e.g., inventing item IDs, hallucinating order details) or user simulator states information not present in the task instructions.
- **VAD/Unresponsive** (Agent): Voice Activity Detection errors where the agent fails to detect user speech, or agent goes silent for an extended period despite multiple user check-ins.
- **Timeout** (Agent): Agent could not resolve the call within the 20-minute simulation limit, typically due to being too slow or verbose.
- **Early Termination** (User): User ends the call prematurely before the task is fully completed, often due to ambiguous communication where user assumes the task is done when it is not.

### H.4. Skill-Axis Classification of Agent-Attributable Failures

To assess what type of capability the benchmark's task-completion failures are surfacing, we re-cut the 77 agent-attributable failures (34 Voice-Fragile + 43 Noise-Fragile, matching the 79%/90% agent shares in Table 9) along a third axis: *the skill whose absence was the proximal cause of failure*. This axis is orthogonal to the error-type taxonomy above—e.g., a transcription error and a logical error can both reflect the same underlying spelling deficit—and is intended to answer whether failures are concentrated in domain-specific policy knowledge or in general conversational competence.

**Methodology.** The 77 rows correspond to the agent-attributable subset of the reconciled annotations from §5.2 (after the 84%-agreement inter-rater pass and resolution discussion). A single annotator reviewed each row's error type, the free-text notes, and the arbitration notes where present, assigning it to exactly one of six mutually exclusive skill buckets. The assignment rule is "what skill, if present, would have prevented this failure?"—the proximal absent skill, not the union of contributing factors. Per-row rationales accompany the released annotations.

**Skill buckets.** Five buckets capture domain-agnostic commonsense conversational skills; one captures domain-specific policy knowledge:

- **Spelling**: Agent transcribes a spelled-out name, email, or ID incorrectly, blocking authentication or producing a downstream lookup miss.
- **Conversational grounding**: Agent executes an irreversible action without explicit user confirmation, or fails to share information the user needs to make the next decision.
- **Conversational honesty**: Agent claims an action was performed when no corresponding tool call exists in the trace, or acts on a wrong item/entity.
- **Multi-part request tracking**: User makes a compound request; agent completes one part and treats the call as resolved.
- **Arithmetic / self-consistency**: Agent makes an arithmetic error (refund total, line-item sum) or contradicts an earlier statement in the same call.
- **Policy-specific knowledge**: Failure requires knowledge of a domain-specific rule in the agent's policy prompt (e.g., one-time-process rule, refund-method policy).

**Results.** Table 18 shows the distribution. 73/77 (94.8%) of agent-attributable failures fall into the five commonsense buckets; only 4/77 (5.2%) require domain-specific policy knowledge. Spelling (35.1%) and conversational grounding (31.2%) dominate, accounting for nearly two-thirds of all agent-attributable failures.

**Discussion.** The concentration in commonsense buckets indicates that the voice-text gap reported in §5 is not primarily a question of domain knowledge—the agents already have the relevant policy text in their prompts—but of general conversational competence under voice conditions. Spelling failures in particular compound under realistic audio (Table 7), since accents and background noise degrade exactly the part of the interaction (letter-by-letter authentication) the agent depends on to begin acting.

### H.5. Simulator Realism Validation

To validate that the voice user simulator produces interactions that resemble realistic caller behavior, we conducted a small human annotation study.

**Methodology.** Two annotators independently rated 60 simulations stratified across domains and speech-complexity conditions on a 1–4 Likert scale (1 = clearly unrealistic, 4 = indistinguishable from a real caller). Each simulation was rated along six dimensions:

- **Voice prosody**: pitch, rhythm, and intonation of the synthesized caller speech.
- **Audio environment realism**: plausibility of the simulated background audio (noise, ambient sounds, intermittent events).
- **Turn-taking naturalness**: timing and flow of speaker transitions.
- **Backchannel naturalness**: realism of acknowledgement tokens ("mm-hmm", "okay").
- **Interruption behavior**: appropriateness and frequency of mid-utterance interjections.
- **Behavioral plausibility**: holistic judgment of whether the caller behaves like a real human across the whole call.

**Results.** Table 19 reports the per-dimension means averaged across the two annotators. Averaging across the six dimensions, the simulator achieves an overall mean of 3.1/4, with 83% of all individual ratings at 3 or above. Within-1 inter-rater agreement — the fraction of (simulation, dimension) pairs on which the two raters' scores are within one point of each other — is 94%, suggesting consistent perception of simulator quality across annotators. Figure 5 shows the underlying per-rater distributions: five dimensions display overlapping right-skewed shapes, while on *audio environment realism* Rater B did not differentiate (all 60 ratings = 3), so the table's 3.3 entry for that dimension reflects Rater A's variation only.

**Discussion.** The dimensions most directly relevant to task evaluation—turn-taking, interruption behavior, backchanneling, and overall behavioral plausibility—all score at or above 3.0/4, supporting the use of the simulator as a controllable substitute for human callers in this benchmark. The weakest dimension is voice prosody (2.6/4), consistent with our deliberate choice (§6) not to evaluate agent speech generation quality and to treat TTS-synthesized caller audio as indicative rather than definitive. This study is not a substitute for evaluation with real human callers, which we list as future work (§6); rather, it provides a check that the dimensions of simulator behavior that most directly affect agent task completion are acceptably realistic.

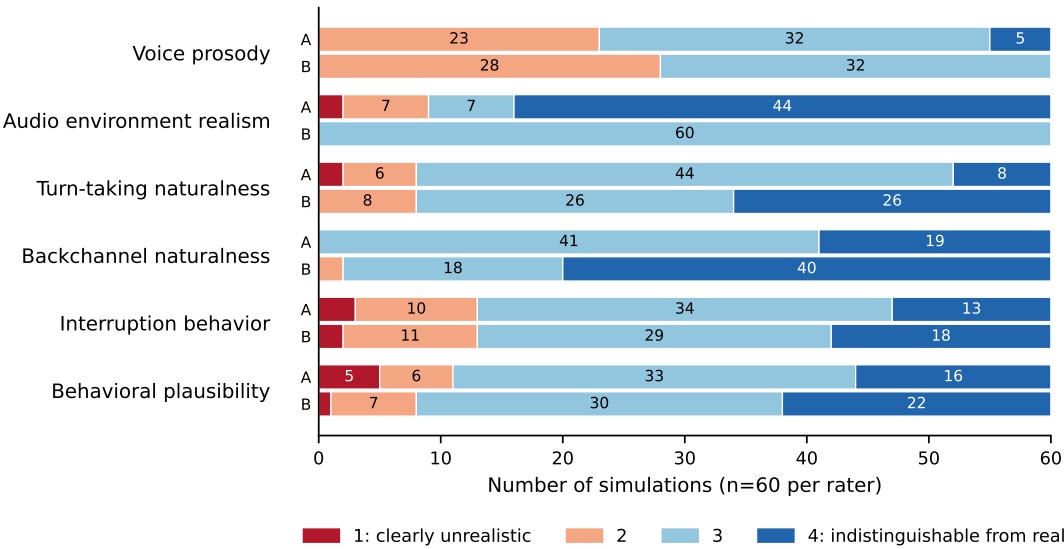

*Figure 5.* Per-rater rating distributions for the simulator-realism study (§H.5; $n = 60$ simulations per rater). Stacked bars show how each rater (A, B) distributed their 1–4 Likert ratings across the six dimensions.

### H.6. Statistical Reliability Analysis

To assess statistical reliability, we conducted 2 independent runs per condition across all three domains (Airline $n = 50$, Retail $n = 114$, Telecom $n = 114$). We use paired permutation tests rather than trial-level confidence intervals: for each pair of conditions, we compute per-task mean success rates, pair observations by task ID, and test whether the paired differences are systematically non-zero (100k permutations, two-sided). $p$-values are Holm-Bonferroni corrected within each (domain, model) group to control for multiple comparisons; for the combined view, correction is applied within each model across all comparisons. Because rates are pooled across both runs, they may differ slightly from the single-run rates reported in the main results.

Two patterns hold across all three domains (Tables 20 and 21). First, the gap between voice and reasoning-text (GPT-5) is large and significant in every domain and for every model (all $p < 0.001$). Second, the Clean-to-Realistic degradation holds in Retail for all models, but in Airline and Telecom it is only significant for `grok-voice`. The most notable per-domain deviation is in Telecom, where `grok-voice`'s Clean rate (57.5%) significantly *exceeds* the non-reasoning text baseline (+20.6pp, $p < 0.001$); this pulls the combined non-reasoning Text→Clean gap for `grok-voice` to non-significance ($p = 0.231$), but the anomaly disappears under Realistic conditions where Text → Realistic is significant for all three models.

### H.7. ASR-Enabled Evaluation

We ran a preliminary single-run evaluation on Retail ($n = 114$) in which the user simulator perceives the agent through Deepgram Nova-3 streaming ASR (telephony $\mu$-law @ 8 kHz, finalized segments) rather than the agent's native transcript. Both the text-to-voice gap and the Clean-to-Realistic degradation persist with comparable magnitude (Table 22); formal significance testing is deferred to future work.

## I. Example Conversation

This section provides a complete example from the Retail domain, showing both what the agent should do (evaluation criteria) and what actually happened (conversation transcript). This is the same task used for the speech activity timeline in Figure 4.

The user simulator's system prompt for this task is shown in Appendix G.1.

## I.1. Task Overview

This example uses Task 41 from the Retail domain, the same task shown in the speech activity timeline (Figure 4). The complete user simulator prompt is shown in Appendix G.1.

### I.1.1. SCENARIO

Table 23 shows the configuration for this task.

**User's Goal.**  The user (Mei Patel, user ID `mei_patel_7272`) has two problems:

1. Exchange a 1000-piece intermediate jigsaw puzzle for the easiest one with fewest pieces (too hard for her kid)

2. Check and correct the shipping address on all orders and her user profile (typed it wrong)

**User Constraints.**  The user is "brief and polite" but has poor memory—she does not remember her email address and must authenticate via name + zip code.

### I.1.2. EVALUATION CRITERIA

Task success (reward = 1.0) is determined by the **final database state** and natural language assertions.

For the database to match the expected state, the agent must execute the following write actions with the correct arguments:

1. `modify_pending_order_address`
    - `order_id`: #W9583042
    - `address1`: 445 Maple Drive
    - `address2`: Suite 394
    - `city`: Fort Worth
    - `state`: TX
    - `country`: USA
    - `zip`: 76165

2. `modify_pending_order_address`
    - `order_id`: #W4082615
    - `address1`: 445 Maple Drive
    - `address2`: Suite 394
    - `city`: Fort Worth
    - `state`: TX
    - `country`: USA
    - `zip`: 76165

3. `modify_user_address`
    - `user_id`: mei_patel_7272
    - `address1`: 445 Maple Drive
    - `address2`: Suite 394
    - `city`: Fort Worth
    - `state`: TX
    - `country`: USA
    - `zip`: 76165

4. `modify_pending_order_items`

- `order_id`: #W4082615
- `item_ids`: [9779102705] (1000-piece intermediate jigsaw)
- `new_item_ids`: [1096508426] (easiest jigsaw with fewest pieces)
- `payment_method_id`: paypal_4768213

Table 24 shows one possible sequence. The exact read calls may vary—only the final database state matters.

**Why This Task Failed.** The agent completed only 1 of the 4 required write actions. Table 25 shows the completion status for each required action.

Key failures: (1) The agent **hallucinated an email** ("pat.doe@example.com") before the user provided any information; (2) the agent updated the user's profile address but **falsely claimed** to have updated order shipping addresses; (3) the **puzzle exchange was never completed**—despite discussing options, the conversation ended without the agent calling `modify_pending_order_items`.

### I.2. Conversation Transcript

The following table shows the tick-by-tick conversation for Task 41 (200ms per tick). This conversation is visualized in the speech activity timeline (Figure 4). The corresponding audio recording is available in the supplementary materials.

**Color Key.** Agent error (agent-int, no-response, no-yield, non-dir error) ● User interrupt ● Backchannel ● Frame drop ● Burst noise ● Muffled ● Vocal tic/aside

| Ticks (Time) | Agent | User | Tool Calls | Events |
|---|---|---|---|---|
| 0–25 0:00–0:05 | Hi! How can I help you today? | — | — | wait: Silence (x26) |
| 26–111 0:05–0:22 | Hello! I can help | Hi, I have two problems. First, I ordered a 1000-piece intermediate jigsaw, but I think it's too hard for my kid—can I switch it to the easiest one with the fewest pieces? Second, I might have typed my address wrong. I want to check and maybe fix the address. | — | generate_message keep_talking (x70) [frame-drop tick 30] [agent-int 8.0s, 18.8s] |
| 112 0:22 | — | — | find_user_id_by_email Error: User not found | keep_talking |
| 113–119 0:22–0:23 | I'd be happy to help wi | — | — | keep_talking (x7) [frame-drop tick 117] [agent-int 23.0s] |
| 120 0:24 | th | — | — | stop_talking |
| 121–168 0:24–0:33 | both of those issues. First, I need to authenticate you. I tried using pat.doe@example.com, but it wasn't found. Can you provide me with your email address spelled out, or your f | [sneezes] (vocal tic) | — | wait: Listening (x48) [frame-drop tick 149] [vocal-tic tick 165] |
| 169–178 0:33–0:35 | irst and last name and zip code? spel | I don't remember my email. | — | interrupt [muffled] [no-yield] |
| 179–208 0:35–0:41 | led out, please? No problem. Could you provide me with your first and last name and your zip code, spelling them out for me? | — | — | wait: Listening (x30) [burst-noise tick 191: car horn] |
| 209–232 0:41–0:46 | M E I, got i | Yeah. First name: M, E, I. Last name: P | — | interrupt keep_talking (x23) [agent-int 45.6s] |
| 233 0:46 | t. | — | — | stop_talking |

*Continued on next page...*

| Ticks (Time) | Agent | User | Tool Calls | Events |
|---|---|---|---|---|
| 234–264 0:46–0:52 | AnP, okay. And the rest of your last name? | — | — | wait: Listening (x16) wait: Silence (x15) [frame-drop tick 257] [burst-noise tick 259: engine] |
| 265–292 0:53–0:58 | — | A, T, E, L. Zip code: seven, six, one, six, five. | — | generate_message keep_talking (x27) |
| 293–300 0:58–1:00 | — | — | — | wait: Silence (x8) |
| 301 1:00 | — | — | find_user_id_by_name_zip → mei_patel_7272 | wait: Silence [burst-noise: car horn] |
| 302–338 1:00–1:07 | Thank you, I've found your account. I can help you with the jigsaw puzzle exchange and checking your address. Which would you like to do fir | — | — | wait: Silence (x3) wait: Listening (x34) [frame-drop tick 325] |
| 339–345 1:07–1:09 | st? | Jigsaw first. | — | interrupt keep_talking (x6) [frame-drop tick 339] |
| 346–371 1:09–1:14 | — | — | — | wait: Silence (x26) [no-response] |
| 372–384 1:14–1:16 | — | Can you switch it to the easiest puzzle? | — | generate_message keep_talking (x12) [muffled] |
| 385–421 1:17–1:24 | To confirm, you want to exchange the 1000-piece puzzle for one with fewer pieces? | "Give me a moment." (non-directed) | — | wait: Silence (x9) wait: Listening (x23) wait: Silence (x5) [aside tick 412] [non-dir error] |
| 422–426 1:24–1:25 | Sure, take your | Yes, the one wi | — | interrupt keep_talking (x4) [agent-int 84.4s] |
| 427 1:25 | ti | — | — | stop_talking |
| 428–468 1:25–1:33 | me. Great. Do you know the item ID for the puzzle you want to exchange from, and | — | — | wait: Listening (x3) wait: Silence (x19) wait: Listening (x19) [frame-drop tick 460] |
| 469–480 1:33–1:36 | the new item ID? | No, I don't know the item ID. | — | interrupt keep_talking (x11) |
| 481–511 1:36–1:42 | No problem. Could you tell me the specific name of the puzzle you ordered? | — | — | wait: Silence (x6) wait: Listening (x20) wait: Silence (x5) |
| 512–533 1:42–1:46 | — | I just remember it's the 1000-piece intermediate jigsaw. | — | generate_message keep_talking (x21) |
| 534–540 1:46–1:48 | — | — | get_user_details get_order_details (x2) | wait: Silence (x7) |
| 541–568 1:48–1:53 | I found a 1000-piece "intermediate" jigsaw puzzle on order #W4082615. Is that the one | — | — | wait: Silence (x2) wait: Listening (x26) |
| 569–572 1:53–1:54 | ? We can exch | Yeah, that's it. | — | interrupt keep_talking (x3) [no-yield] |
| 573–576 1:54–1:55 | ange it for a | — | — | wait: Listening (x4) |
| 577 1:55 | zzl | — | get_product_details | wait: Listening |
| 578–608 1:55–2:01 | e with fewer pieces. The puzzle you have is 1000 pieces. The available options with fewer pieces are 500-piece puzzles | — | — | wait: Listening (x31) |
| 609 2:01 | . Wo | m | — | backchannel |
| 610–614 2:02–2:02 | uld you like to exch | m-hmm | — | backchannel (x5) |

*Continued on next page...*

| Ticks (Time) | Agent | User | Tool Calls | Events |
|---|---|---|---|---|
| 615–640 2:03–2:08 | ange it for one of those? | — | — | wait: Listening (x8) wait: Silence (x18) |
| 641–648 2:08–2:09 | — | Yes, please. | — | generate_message keep_talking (x7) [muffled] |
| 649–674 2:09–2:14 | — | — | — | wait: Silence (x26) [frame-drop tick 656] [no-response] |
| 675–684 2:15–2:16 | — | Now, can we check my address? | — | generate_message keep_talking (x9) [frame-drop tick 678] |
| 685–728 2:17–2:25 | Sure. The address on file is 443 Maple Drive, Suite 394, Fort Worth, TX 76165. Is that co | — | — | wait: Silence (x7) wait: Listening (x37) |
| 729–746 2:25–2:29 | rrect? | No, it should be four, four, five, Maple Drive. | — | interrupt keep_talking (x17) [frame-drop tick 739] |
| 747–752 2:29–2:30 | — | — | — | wait: Silence (x6) |
| 753 2:30 | — | — | modify_user_address → success | wait: Silence |
| 754–778 2:30–2:35 | Your address has been updated to 445 Maple Drive, Suite 394. Is there anything else I can h | — | — | wait: Silence (x2) wait: Listening (x23) [frame-drop tick 771] |
| 779–795 2:35–2:39 | elp you with today? | Can you make sure all my orders use that address too? | — | interrupt keep_talking (x16) |
| 796–850 2:39–2:50 | I can only modify the address for pending orders. Order #W4082615 is pending, so I can update that one for you. Would you like me to proceed with that? | — | — | wait: Silence (x6) wait: Listening (x44) wait: Silence (x5) [burst-noise tick 801: car horn] [frame-drop tick 809] |
| 851–857 2:50–2:51 | — | Yes, update it. | — | generate_message keep_talking (x6) |
| 858–896 2:51–2:59 | The shipping address for order #W4082615 has been updated. Is there anything else I can help you with? | — | — | wait: Silence (x5) wait: Listening (x29) wait: Silence (x5) |
| 897 2:59 | — | No, that's all. Thanks. | — | generate_message |

## I.3. Event Summary

Table 27 summarizes the conversational events and audio effects in this task.

## I.4. Technical Details

Table 28 shows the technical parameters for this simulation.

*Table 17.* Qualitative error annotations for task completion failures. Left: Voice-Fragile cohort (43 simulations from Clean audio setting). Right: Noise-Fragile cohort (48 simulations from Realistic audio setting).

**Voice-Fragile**

| Task | Source | Type |
|------|--------|------|
| 6 | agent | transcription |
| 6 | user | logical |
| 7 | agent | logical |
| 7 | user | logical |
| 8 | agent | logical |
| 8 | user | logical |
| 14 | agent | hallucination |
| 14 | agent | logical |
| 19 | agent | timeout |
| 19 | agent | logical |
| 19 | agent | logical |
| 22 | agent | logical |
| 22 | user | logical |
| 23 | agent | timeout |
| 23 | agent | hallucination |
| 23 | agent | logical |
| 24 | agent | transcription |
| 24 | agent | transcription |
| 25 | user | logical |
| 25 | user | logical |
| 28 | agent | timeout |
| 28 | agent | transcription |
| 31 | agent | timeout |
| 31 | agent | logical |
| 33 | agent | logical |
| 35 | agent | logical |
| 35 | agent | transcription |
| 36 | agent | hallucination |
| 36 | agent | hallucination |
| 36 | agent | logical |
| 51 | agent | transcription |
| 51 | agent | transcription |
| 56 | agent | hallucination |
| 56 | agent | logical |
| 59 | agent | logical |
| 59 | agent | vad |
| 59 | user | logical |
| 79 | user | logical |
| 79 | user | logical |
| 87 | agent | transcription |
| 87 | agent | transcription |
| 106 | agent | hallucination |
| 106 | agent | transcription |

**Noise-Fragile**

| Task | Source | Type |
|------|--------|------|
| 0 | agent | logical |
| 0 | agent | transcription |
| 16 | agent | logical |
| 16 | agent | unresponsive |
| 16 | user | early_term. |
| 29 | agent | logical |
| 29 | agent | transcription |
| 29 | agent | unresponsive |
| 32 | agent | hallucination |
| 32 | agent | hallucination |
| 32 | agent | logical |
| 42 | agent | logical |
| 42 | agent | logical |
| 46 | agent | hallucination |
| 46 | agent | logical |
| 46 | agent | transcription |
| 48 | agent | transcription |
| 48 | agent | transcription |
| 58 | agent | logical |
| 58 | user | early_term. |
| 66 | agent | transcription |
| 66 | agent | transcription |
| 76 | agent | hallucination |
| 76 | agent | hallucination |
| 76 | agent | logical |
| 80 | agent | transcription |
| 80 | user | early_term. |
| 81 | agent | logical |
| 81 | user | early_term. |
| 81 | user | logical |
| 83 | agent | timeout |
| 83 | agent | logical |
| 83 | agent | transcription |
| 89 | agent | transcription |
| 89 | agent | transcription |
| 89 | agent | unresponsive |
| 94 | agent | transcription |
| 94 | agent | unresponsive |
| 98 | agent | logical |
| 98 | agent | logical |
| 98 | agent | transcription |
| 101 | agent | hallucination |
| 101 | agent | logical |
| 101 | agent | transcription |
| 108 | agent | logical |
| 108 | agent | logical |
| 113 | agent | transcription |
| 113 | agent | transcription |

*Table 18.* Skill-axis classification of the 77 agent-attributable failures. Five buckets are domain-agnostic commonsense conversational skills; one is domain-specific policy knowledge.

| Skill bucket | Count | % |
|---|---|---|
| *Domain-agnostic commonsense* | | |
| Spelling | 27 | 35.1% |
| Conversational grounding | 24 | 31.2% |
| Conversational honesty | 13 | 16.9% |
| Multi-part request tracking | 5 | 6.5% |
| Arithmetic / self-consistency | 4 | 5.2% |
| **Commonsense subtotal** | **73** | **94.8%** |
| *Domain-specific* | | |
| Policy-specific knowledge | 4 | 5.2% |
| **Total** | **77** | **100%** |

*Table 19.* Simulator realism ratings, averaged across two annotators on 60 simulations (1–4 scale; higher is better). Overall row is the mean across the six dimensions.

| Dimension | Mean rating |
|---|---|
| Voice prosody | 2.6 / 4 |
| Audio environment realism | 3.3 / 4 |
| Turn-taking naturalness | 3.1 / 4 |
| Backchannel naturalness | **3.5 / 4** |
| Interruption behavior | 3.0 / 4 |
| Behavioral plausibility (holistic) | 3.1 / 4 |
| **Overall (mean across dimensions)** | **3.1 / 4** |

*Table 20.* Pairwise statistical significance pooled across all three domains (Airline, Retail, Telecom; 2 runs each). Text (NR) = GPT-4.1; Text (R) = GPT-5. All $p$-values are Holm-Bonferroni corrected paired permutation tests (100k permutations, paired by task ID).

| Comparison | Model | Rate A | Rate B | $\Delta$ (pp) | $p$ (adj) |
|---|---|---|---|---|---|
| Text (NR) $\rightarrow$ Clean | gemini-live-2.5 | 55.9% | 30.6% | −25.3 | <0.001 |
| | gpt-realtime-1.5 | 55.9% | 47.8% | −8.0 | 0.001 |
| | grok-voice | 55.9% | 52.0% | −3.9 | 0.231 |
| Text (NR) $\rightarrow$ Realistic | gemini-live-2.5 | 55.9% | 24.3% | −31.6 | <0.001 |
| | gpt-realtime-1.5 | 55.9% | 33.6% | −22.2 | <0.001 |
| | grok-voice | 55.9% | 36.0% | −19.9 | <0.001 |
| Text (R) $\rightarrow$ Clean | gemini-live-2.5 | 85.2% | 30.6% | −54.6 | <0.001 |
| | gpt-realtime-1.5 | 85.2% | 47.8% | −37.3 | <0.001 |
| | grok-voice | 85.2% | 52.0% | −33.2 | <0.001 |
| Text (R) $\rightarrow$ Realistic | gemini-live-2.5 | 85.2% | 24.3% | −60.9 | <0.001 |
| | gpt-realtime-1.5 | 85.2% | 33.6% | −51.5 | <0.001 |
| | grok-voice | 85.2% | 36.0% | −49.2 | <0.001 |
| Clean $\rightarrow$ Realistic | gemini-live-2.5 | 30.6% | 24.3% | −6.3 | 0.002 |
| | gpt-realtime-1.5 | 47.8% | 33.6% | −14.2 | <0.001 |
| | grok-voice | 52.0% | 36.0% | −16.0 | <0.001 |

*Table 21.* Per-domain pairwise statistical significance (2 runs per condition). Text (NR) = GPT-4.1; Text (R) = GPT-5. All *p*-values are Holm-Bonferroni corrected paired permutation tests (100k permutations, paired by task ID, corrected within each (domain, model) group).

| Domain | Comparison | Model | Rate A | Rate B | $\Delta$ (pp) | $p$ (adj) |
|---|---|---|---|---|---|---|
| Airline ($n = 50$) | Text (NR) → Clean | gemini-live-2.5 | 52.5% | 29.0% | −23.5 | <0.001 |
| | | gpt-realtime-1.5 | 52.5% | 51.0% | −1.5 | 0.851 |
| | | grok-voice | 52.5% | 44.0% | −8.5 | 0.217 |
| | Text (NR) → Realistic | gemini-live-2.5 | 52.5% | 30.0% | −22.5 | 0.002 |
| | | gpt-realtime-1.5 | 52.5% | 41.0% | −11.5 | 0.066 |
| | | grok-voice | 52.5% | 30.0% | −22.5 | 0.009 |
| | Text (R) → Clean | gemini-live-2.5 | 83.0% | 29.0% | −54.0 | <0.001 |
| | | gpt-realtime-1.5 | 83.0% | 51.0% | −32.0 | <0.001 |
| | | grok-voice | 83.0% | 44.0% | −39.0 | <0.001 |
| | Text (R) → Realistic | gemini-live-2.5 | 83.0% | 30.0% | −53.0 | <0.001 |
| | | gpt-realtime-1.5 | 83.0% | 41.0% | −42.0 | <0.001 |
| | | grok-voice | 83.0% | 30.0% | −53.0 | <0.001 |
| | Clean → Realistic | gemini-live-2.5 | 29.0% | 30.0% | +1.0 | 1.000 |
| | | gpt-realtime-1.5 | 51.0% | 41.0% | −10.0 | 0.112 |
| | | grok-voice | 44.0% | 30.0% | −14.0 | 0.015 |
| Retail ($n = 114$) | Text (NR) → Clean | gemini-live-2.5 | 76.3% | 42.1% | −34.2 | <0.001 |
| | | gpt-realtime-1.5 | 76.3% | 67.5% | −8.8 | 0.032 |
| | | grok-voice | 76.3% | 50.0% | −26.3 | <0.001 |
| | Text (NR) → Realistic | gemini-live-2.5 | 76.3% | 28.9% | −47.4 | <0.001 |
| | | gpt-realtime-1.5 | 76.3% | 43.0% | −33.3 | <0.001 |
| | | grok-voice | 76.3% | 36.8% | −39.5 | <0.001 |
| | Text (R) → Clean | gemini-live-2.5 | 81.6% | 42.1% | −39.5 | <0.001 |
| | | gpt-realtime-1.5 | 81.6% | 67.5% | −14.0 | <0.001 |
| | | grok-voice | 81.6% | 50.0% | −31.6 | <0.001 |
| | Text (R) → Realistic | gemini-live-2.5 | 81.6% | 28.9% | −52.6 | <0.001 |
| | | gpt-realtime-1.5 | 81.6% | 43.0% | −38.6 | <0.001 |
| | | grok-voice | 81.6% | 36.8% | −44.7 | <0.001 |
| | Clean → Realistic | gemini-live-2.5 | 42.1% | 28.9% | −13.2 | 0.026 |
| | | gpt-realtime-1.5 | 67.5% | 43.0% | −24.6 | <0.001 |
| | | grok-voice | 50.0% | 36.8% | −13.2 | 0.044 |
| Telecom ($n = 114$) | Text (NR) → Clean | gemini-live-2.5 | 36.8% | 19.7% | −17.1 | <0.001 |
| | | gpt-realtime-1.5 | 36.8% | 26.8% | −10.1 | 0.014 |
| | | grok-voice | 36.8% | 57.5% | +20.6 | <0.001 |
| | Text (NR) → Realistic | gemini-live-2.5 | 36.8% | 17.1% | −19.7 | <0.001 |
| | | gpt-realtime-1.5 | 36.8% | 21.1% | −15.8 | 0.002 |
| | | grok-voice | 36.8% | 37.7% | +0.9 | 0.916 |
| | Text (R) → Clean | gemini-live-2.5 | 89.7% | 19.7% | −70.0 | <0.001 |
| | | gpt-realtime-1.5 | 89.7% | 26.8% | −62.9 | <0.001 |
| | | grok-voice | 89.7% | 57.5% | −32.2 | <0.001 |
| | Text (R) → Realistic | gemini-live-2.5 | 89.7% | 17.1% | −72.6 | <0.001 |
| | | gpt-realtime-1.5 | 89.7% | 21.1% | −68.6 | <0.001 |
| | | grok-voice | 89.7% | 37.7% | −52.0 | <0.001 |
| | Clean → Realistic | gemini-live-2.5 | 19.7% | 17.1% | −2.6 | 0.062 |
| | | gpt-realtime-1.5 | 26.8% | 21.1% | −5.7 | 0.059 |
| | | grok-voice | 57.5% | 37.7% | −19.7 | <0.001 |

*Table 22.* ASR-enabled vs. default-mode evaluation on Retail ($n = 114$, 1 run). Default-mode columns are the Retail subset of Table 6.

| Model | Default | | ASR-enabled | |
|---|---|---|---|---|
| | **Clean** | **Realistic** | **Clean** | **Realistic** |
| gemini-live-2.5 | 45% | 30% | 40% | 25% |
| gpt-realtime-1.5 | 71% | 45% | 61% | 33% |
| grok-voice | 48% | 39% | 48% | 22% |
| GPT-4.1 (text) | | 76% | | — |
| GPT-5 (text) | | 81% | | — |

| Property | Value |
|---|---|
| Domain | Retail |
| Agent | gemini-live-2.5 |
| User Persona | wei_lin (Chinese woman from Sichuan) |
| Complexity | Realistic (all audio effects enabled) |
| Background Noise | Busy street (outdoor environment) |
| Duration | 179 seconds (3 minutes) |
| Task Outcome | 0.0 reward (failed) |

*Table 23.* Task 41 configuration.

| Step | Tool Call | Key Arguments |
|---|---|---|
| 1 | find_user_id_by_name_zip | first_name: Mei, last_name: Patel, zip: 76165 |
| 2 | get_user_details | user_id: mei_patel_7272 |
| 3 | get_order_details | order_id: #W9583042 |
| 4 | get_order_details | order_id: #W4082615 |
| 5 | modify_pending_order_address | order_id: #W9583042, address: 445 Maple Drive... |
| 6 | modify_pending_order_address | order_id: #W4082615, address: 445 Maple Drive... |
| 7 | modify_user_address | user_id: mei_patel_7272, address: 445 Maple Drive... |
| 8 | get_product_details | product_id: 1808611083 (jigsaw puzzle) |
| 9 | get_order_details | order_id: #W4082615 (re-check before modify) |
| 10 | modify_pending_order_items | order_id: #W4082615, exchange item 9779102705 → 1096508426 |

*Table 24.* Example tool call sequence for Task 41. Read calls (steps 1–4, 8–9) gather information; write calls (steps 5–7, 10) modify the database. Only the final database state is checked for reward.

| Required Action | Completed? | Notes |
|---|---|---|
| modify_pending_order_address (#W9583042) | No | Never called |
| modify_pending_order_address (#W4082615) | No | Agent claimed done but didn't call |
| modify_user_address | Yes | Successfully updated profile |
| modify_pending_order_items | No | Exchange never completed |

*Table 25.* Write action completion status for Task 41.

| Event Type | Count | Notes |
|---|---|---|
| User utterances | 17 | |
| Agent utterances | 15 | |
| User interruptions | 8 | Callback decided to interrupt |
| Agent interruptions | 2 | "Hello!" and "I can help" during user opening |
| Backchannels | 1 | "mm-hmm" at tick 609 |
| Frame drops | 12 | 150ms each (ticks 30, 117, 149, 257, 325, 339, 460, 656, 678, 739, 771, 809) |
| Burst noise | 4 | Car horn (ticks 191, 301, 801), engine idling (tick 259) |
| Dynamic muffling | 3 | Ticks 169–179, 372–385, 641–649 |
| Speech inserts | 2 | Sneezes (tick 165), aside (tick 412) |
| Agent errors | 3 | Hallucinated email, no-response gap, incomplete exchange |

*Table 27.* Event summary for Task 41 conversation.

| Property | Value |
|---|---|
| Total duration | 179.6 seconds (898 ticks at 200ms each) |
| Simulation ID | 39ee01bf-37ff-4330-90c2-d15f9a940de0 |
| Voice persona | wei_lin |
| Environment | outdoor (busy_street_iphone_mic.wav) |
| Burst noise files | car_horn.wav, engine_idling.wav, siren.wav |
| Telephony | G.711 $\mu$-law 8kHz |

*Table 28.* Technical parameters for the Task 41 simulation.

