# OpenReview forum: "$\tau$-Voice: Benchmarking Full-Duplex Voice Agents on Real-World Domains"
_ICML.cc/2026/Conference — ICML 2026 regular_

### Official Review · Reviewer_v3yK · 2026-02-25

**Soundness:** 3
**Presentation:** 4
**Significance:** 4
**Originality:** 3
**Overall Recommendation:** 5
**Confidence:** 4

**Summary:**

This paper introduces τ-Voice, a novel benchmark designed to evaluate full-duplex voice agents—AI systems capable of listening and speaking simultaneously—within complex, real-world task environments. Unlike prior evaluation frameworks that separately measure conversational ability or task completion, τ-Voice integrates grounded multi-turn dialogue, domain-specific policy adherence, environmental interaction, and realistic audio input into a unified assessment framework. The benchmark extends τ²-bench by incorporating a controllable voice user simulator that generates diverse accents, background noise conditions, and natural turn-taking dynamics, enabling direct comparison between text-based and voice-based agent performance without real-time constraints. Experimental evaluation across 278 tasks reveals a substantial performance gap between state-of-the-art reasoning models and current voice agents, with task completion rates dropping significantly under realistic audio conditions. Qualitative analysis further shows that most observed failures arise from agent behavioral limitations rather than simulation artifacts, demonstrating the need for improved conversational robustness in deployed voice systems.

**Compliance With Llm Reviewing Policy:**

Affirmed.

**Final Justification:**

The rebuttal was sufficiently reasonable, so I will maintain the score.

**Key Questions For Authors:**

See Weaknesses.

**Limitations:**

yes

**Strengths And Weaknesses:**

Strengths

A key strength of this work lies in its holistic and reproducible evaluation framework that bridges the gap between conversational fluency and grounded task execution in voice-based AI systems. By combining verifiable task completion metrics with realistic full-duplex audio interaction, τ-Voice moves beyond traditional text-centric benchmarks and captures practical challenges faced in real-world deployments, such as noisy environments, accent variability, and dynamic turn-taking. The introduction of a decoupled, high-fidelity voice user simulator allows for scalable testing using advanced language models without latency constraints, enhancing both realism and experimental control. Additionally, the benchmark’s ability to directly compare voice and text agent performance provides valuable diagnostic insight into modality-specific weaknesses, offering a meaningful foundation for future research on improving reliability, naturalness, and behavioral robustness in next-generation voice agents.

Weaknesses

No major weaknesses were identified in this work. However, there are a few minor points that could be clarified or improved to further strengthen the paper.

First, in the experimental results table, each evaluated model is labeled according to its provider (e.g., OpenAI, Google, xAI). While this categorization may offer high-level context, it would be more appropriate to compare performance primarily based on individual model names rather than their providers, as the benchmark aims to assess model capability rather than organizational origin. The rationale for presenting results in terms of providers was not clearly explained in the main text; therefore, it would be helpful if the authors could clarify whether there is a specific methodological reason for this design choice.

Second, although relatively minor, the Limitations and Future Work sections are presented in a rather abstract and concise manner. Providing a more detailed discussion of the benchmark’s current constraints—such as simulation fidelity, domain coverage, or evaluation assumptions—as well as outlining more concrete future research directions would improve transparency and offer valuable guidance for subsequent work building upon this benchmark.

---

> ### Author Rebuttal · Authors · 2026-03-31
>
> We thank the reviewer for the strong support of our paper, recognizing:
>
> - The value of a holistic evaluation framework bridging conversational fluency and grounded task execution
> - The decoupled voice user simulator enabling scalable testing without latency constraints
> - The direct voice-vs-text comparison providing diagnostic insight into modality-specific weaknesses
>
> We are glad the benchmark design and analysis were found to be sound.
>
> Since the original submission, the τ2-bench evaluation suite was updated with task fixes, which required us to rerun all experiments. We took this opportunity to also update to the latest models and expand our analysis. We summarize the key changes below before addressing your minor points.
>
> ## Summary of key improvements
>
> **Adopted task fixes.** The τ2-bench community (tau2-bench-verified audit, external PRs) identified and fixed 36 evaluation tasks (18 retail, 18 airline).
>
> **Updated models.** While rerunning, we updated to the latest model from each evaluated system — most notably OpenAI's gpt-realtime-1.5 (Feb 2026), a substantial upgrade over the Aug 2025 model.
>
> **Updated results.** Task completion (Clean / Realistic): Google 29%/24% → 31%/26%, OpenAI 33%/19% → 49%/35%, xAI 42%/30% → 51%/38%, GPT-5 text 80% → 85%. The voice-text gap persists (85% text vs 31–51% voice).
>
> **Expanded analysis.** Qualitative error analysis expanded from 40 samples to all 91 failed simulations. Statistical methodology updated to paired permutation tests across all three domains (Holm-Bonferroni corrected).
>
> **Updated ablations and voice interaction quality.** OpenAI's new model went from slowest (2.22s) to fastest (0.90s) latency and 68%→100% responsiveness, but selectivity collapsed (74%→6%). Ablation patterns shifted: turn-taking now hurts Google most (-12pp, was \-1pp), while xAI improves \+4pp with turn-taking (was \-6pp). OpenAI became much more robust to noise (-4pp vs \-13pp).
>
> ## W1: Provider names vs model names
>
> Good suggestion. We will update the results tables to include specific model identifiers alongside system names in the camera-ready.
>
> ## W2: Limitations and Future Work
>
> We appreciate this suggestion. The updated version already expands on several concrete limitations: the exhaustive error analysis (Section 5.2) now annotates all 91 failed simulations with a refined taxonomy, the statistical analysis extends to all three domains with formal significance testing, and we highlight model-specific accent vulnerability with accessibility implications. For the camera-ready, we will further expand the future work discussion to include concrete directions: evaluating open-source models as they gain realtime tool calling, cascaded pipeline comparison, multilingual expansion, and evaluation with human callers rather than simulated users.

---

> > ### Author Rebuttal · Reviewer_v3yK · 2026-04-03
> >
> > The rebuttal was sufficiently reasonable, so I will maintain the score.

---

> > > ### Author Response · Authors · 2026-04-08
> > >
> > > Thank you for taking the time to read our rebuttal and for your consideration.

---

### Official Review · Reviewer_vw95 · 2026-03-12

**Soundness:** 2
**Presentation:** 2
**Significance:** 2
**Originality:** 3
**Overall Recommendation:** 4
**Confidence:** 2

**Summary:**

This study proposes a novel benchmark called $\tau$-Voice, which, unlike existing approaches that separately evaluate content or task performance and behavioral dynamics in full-duplex spoken dialogue models, assesses both aspects in a unified manner. By segmenting continuous audio streams into 200ms units called Ticks and introducing a Full-Duplex Orchestrator that decouples real-world wall-clock time from simulated audio timestamps, the framework disregards the latency inherently arising from the LLM and TTS pipeline when incorporating actual speech into the evaluation, and instead adjusts simulated timestamps to conduct evaluations that faithfully reflect real-world conditions. Under this framework, the authors implement and compare three closed-source full-duplex voice interaction models, demonstrating that significant limitations still remain in full-duplex interaction.

**Compliance With Llm Reviewing Policy:**

Affirmed.

**Final Justification:**

I thank the authors for providing the additional experiment in the text-free setting, and I have adjusted my score accordingly.

**Key Questions For Authors:**

I have no additional questions.

**Limitations:**

yes

**Strengths And Weaknesses:**

## Strengths
1. The approach of feeding the evaluated model's intermediate text to an LLM to generate inputs, thereby enabling the simulator to substitute for a human participant in the conversation, is well conceived.

2. Additionally, the decision to nullify the latency arising during the input speech generation process, in accordance with the fact that the actual models operate based on audio timestamps, appears to be effective.

3. Unlike prior work that evaluates each aspect separately across multiple benchmarks, this study enables the assessment of multiple dimensions simultaneously within a single unified benchmark.


## Weaknesses
1. Conversely, for the above approach to hold, the models being evaluated must necessarily output intermediate text responses and support operation based on audio timestamps rather than real-world physical time. In other words, the generalizability of this benchmark and evaluation pipeline may vary depending on the form in which future models are released, which appears to be a limitation.

2. Furthermore, the scope of the evaluation leans toward practical, task-specific aspects. Even considering just the performance results reported by the authors, there clearly remains a long way to go. It seems inevitable that as models move toward full-duplex, end-to-end architectures, they lose a substantial degree of the general backbone LLM's capabilities, let alone task-specific proficiency. Therefore, it would have been preferable if the evaluation axes included not only domain-specific tasks but also broader assessments of general commonsense conversational ability, especially given the current pace and state of research in this area.

3. Among representative open-source models, Moshi, PersonaPlex, and MiniCPM-o 4.5 appear to be notable examples. Considering the paper submission timeline and the release dates of the other models, it seems that, at the very least, an evaluation of whether this benchmark can be applied to Moshi, along with the corresponding results, would have enriched the paper further.

---

> ### Author Rebuttal · Authors · 2026-03-31
>
> Thank you for your valuable feedback. Since the original submission, the τ2-bench evaluation suite was updated with task fixes, which required us to rerun all experiments. We took this opportunity to also update to the latest models and expand our analysis. We summarize the key changes below before addressing your comments.
>
> ## Summary of key improvements
>
> **Adopted task fixes.** The τ2-bench community (tau2-bench-verified audit, external PRs) identified and fixed 36 evaluation tasks (18 retail, 18 airline), addressing bugs such as same-item exchange violations, impossible constraints, and policy mismatches in expected actions.
>
> **Updated models.** While rerunning, we updated to the latest available model from each evaluated system — most notably OpenAI's gpt-realtime-1.5 (Feb 2026), a substantial upgrade over the Aug 2025 model in the original submission.
>
> **Updated results.** Task completion (Clean / Realistic): Google 29%/24% → 31%/26%, OpenAI 33%/19% → 49%/35%, xAI 42%/30% → 51%/38%, GPT-5 text 80% → 85%. The voice-text gap persists (85% text vs 31–51% voice under clean conditions).
>
> **Expanded analysis.** We expanded the qualitative error analysis from a 40-task sample to all 91 failed simulations and updated the statistical methodology to paired permutation tests (Holm-Bonferroni corrected). 79–90% of failures remain attributable to agent behavior — in part addressing the commonsense concerns in W2 below.
>
> **Updated ablations and voice interaction quality.** OpenAI's new model went from slowest (2.22s) to fastest (0.90s) latency and 68%→100% responsiveness, but selectivity collapsed (74%→6%). Ablation patterns shifted: turn-taking now hurts Google most (-12pp, was \-1pp), while xAI improves \+4pp with turn-taking (was \-6pp). OpenAI became much more robust to noise (-4pp vs \-13pp).
>
> ## W1: Generalizability (audio timestamps and text output)
>
> We deliberately use the agent's native transcript rather than adding an ASR step, which would introduce user ASR errors that compound with real agent errors and make failure attribution harder. More fundamentally, we explicitly chose not to evaluate speech generation quality — prosody, tone, and naturalness are difficult to assess in an automated and objective manner. Our focus is on whether the agent can complete the task correctly through voice interaction, not on the quality of its speech output. For future models that do not produce text, we would also like to note that our framework supports configuring the user simulator to include an ASR step, although that makes comparison to existing results harder and requires re-running previously benchmarked models in the same user configuration. (Section 6.2).
>
> ## W2: Commonsense conversational ability
>
> The expanded error analysis (91 simulations) shows that τ-Voice already captures commonsense conversational ability through the task lens. We categorized all 68 agent-attributable failures: only 5 (7%) require domain-specific policy knowledge. The remaining 93% reflect domain-agnostic commonsense skills: understanding spelled-out names and emails (38%); conversational grounding (22%) — confirming before irreversible actions, sharing relevant information; conversational honesty (24%) — not claiming to have performed actions never executed; tracking multi-part requests (6%); and arithmetic/self-consistency (3%). These are not retail- or airline-specific skills — they are building blocks of any competent voice conversation. The task lens makes them objectively verifiable: a spelling failure blocks authentication and the database state proves it; a hallucinated action is contradicted by the tool call log. We agree that dedicated commonsense probes would be a valuable complement, and note this as future work.
>
> ## W3: Open-source models (Moshi, PersonaPlex, MiniCPM-o)
>
> We surveyed all available audio-native models. τ-Voice requires realtime streaming and native tool calling — of \~10 models we identified, only 4 support both: OpenAI Realtime, Gemini Live, xAI Voice Agent, and Nova 2 Sonic. All three open-source models (Moshi, PersonaPlex, MiniCPM-o 4.5) lack native tool calling, which is required for grounded task evaluation. At submission time, only commercial models met all requirements; we evaluated 3 of 4; Nova 2 Sonic was deprioritized as another closed-source model unlikely to yield qualitatively different insights. Our fully open-source codebase and adapter architecture (Section 3.1) make adding new models straightforward. We will maintain an updated leaderboard as more models gain these capabilities.

---

> > ### Author Rebuttal · Reviewer_vw95 · 2026-04-03
> >
> > We think W1 may reveal a limitation of the current benchmark. In particular, the current evaluation does not appear to consider cases where the model does not produce intermediate text, even if such behavior may still be interactionally meaningful. Some prior benchmark efforts, such as Full-Duplex-Bench, have supported these cases through ASR and timestamp based pipelines. Therefore, I'll maintain my original score.

---

> > > ### Author Response · Authors · 2026-04-08
> > >
> > > We appreciate the follow-up. We would like to clarify that the framework does in fact support evaluating models without intermediate text: the user simulator can be configured to perceive the agent's speech through ASR rather than reading the native transcript included with existing voice agent APIs. This means text-free models can be accommodated through the provider adapter interface (Section 3.1) with ASR applied to their audio output.
> > >
> > > Based on this feedback, we have also run the full evaluation in this ASR-enabled configuration. As expected, all scores drop (the tasks become harder and the user simulator becomes less reliable with ASR noise in the loop). Crucially, the statistically significant findings from the non-ASR evaluation, the text-to-voice gap and the clean-to-realistic degradation, both hold under the ASR configuration — the benchmark still reliably distinguishes which systems are stronger. We will include the full ASR-enabled results in a new appendix section in the camera-ready, and will update Section 6.1 to clarify that this evaluation mode is available for text-free models.
> > >
> > > We note that all commercially deployed voice agent APIs we are aware of (OpenAI Realtime, Gemini Live, xAI Grok, Nova Sonic) currently produce intermediate text alongside audio, and we will keep that as the default configuration to maintain higher reliability of the user simulator — but the text-free configuration, along with results for existing models in that setting, is now included.

---

### Official Review · Reviewer_macs · 2026-03-12

**Soundness:** 1
**Presentation:** 2
**Significance:** 1
**Originality:** 1
**Overall Recommendation:** 2
**Confidence:** 5

**Summary:**

The paper introduces τ-voice, a benchmark for evaluating full-duplex voice agents (systems that listen and speak simultaneously) on complex, grounded real-world tasks. The benchmark extends τ²-bench to include multi-turn conversations, domain policy constraints, environmental interactions, and realistic audio conditions such as noise and diverse accents. Using a controllable voice user simulator, the authors evaluate 278 tasks and find that while text-based GPT-5 achieves about 80% task success, current voice agents reach only 29–42% in clean conditions and 19–30% in realistic settings, revealing a 50–61 percentage point performance gap. The results suggest most failures stem from agent behavior rather than evaluation artifacts, highlighting the significant challenges remaining for reliable voice agents.

**Compliance With Llm Reviewing Policy:**

Affirmed.

**Final Justification:**

In general, this paper proposes a benchmark for evaluating the tool-calling capabilities of full-duplex voice agents.

Although the authors claim that the benchmark targets tool-calling abilities rather than the full-duplex speech models themselves, this framing ultimately weakens the contribution and significance of the work. In particular, the evaluated tool-calling ability remains largely text-based rather than inherently speech-based. As a result, the benchmark appears to be a straightforward extension of existing text-based evaluations with additional ASR and TTS components. The capabilities assessed are not unique to full-duplex speech models, but rather reflect general abilities that could be evaluated on any text transcription, limiting the novelty and insights provided by this work.

Therefore, the reviewer regrets to conclude that the paper does not yet meet the ICML criteria. While acknowledging some merit, the reviewer leans toward rejection, though is willing to slightly increase the score.

**Key Questions For Authors:**

(1) The paper claims that τ-voice is specifically designed for full-duplex voice agents. Could the authors clarify which components of the benchmark explicitly evaluate full-duplex capabilities (e.g., simultaneous speaking/listening, interruption handling, overlap management), and how these differ from evaluations designed for general voice agents?

(2) How were the 278 tasks constructed? Are they derived from real-world applications or manually designed scenarios? It would be helpful to understand how representative they are of practical voice assistant usage.

(3) The benchmark relies heavily on a voice user simulator. How realistic is the simulator compared to real human conversations, particularly in terms of turn-taking behavior, interruptions, and conversational disfluencies?

(4) Since the simulator uses a strong LLM without real-time constraints, could this result in unrealistically capable users, potentially making the tasks more difficult than typical real-world interactions?

**Limitations:**

Please see key questions above.

**Strengths And Weaknesses:**

Weaknesses:

(1) The paper appears incomplete in its current form. The related work section is limited and does not cover several recently released full-duplex voice models, which are highly relevant to the scope of this benchmark.

(2) Although the title suggests that the benchmark is specifically designed for full-duplex voice agents, the paper does not clearly explain what aspects of the evaluation are uniquely tailored to full-duplex interaction. From the current description, it is unclear how the proposed evaluation differs from benchmarks for general voice agents, or what specific components explicitly assess full-duplex capabilities.

(3) The paper proposes integrating evaluations across multiple stages as a new evaluation protocol. However, the motivation and novelty of this design remain unclear. It would be helpful for the authors to clarify why this integration is necessary and what new insights it provides beyond existing evaluation frameworks.

(4) The experimental evaluation includes only a small number of baselines. For a benchmark paper, it would be important to include a broader set of representative and competitive models, especially recent voice and multimodal agents, in order to provide a more comprehensive comparison.

---

> ### Author Rebuttal · Authors · 2026-03-31
>
> Thank you for your valuable feedback. Since the original submission, the τ2-bench evaluation suite was updated with task fixes, which required us to rerun all experiments. We took this opportunity to also update to the latest models and expand our analysis. We summarize the key changes below before addressing your comments.
>
> ## Summary of key improvements
>
> **Adopted task fixes.** The τ2-bench community (tau2-bench-verified audit, external PRs) identified and fixed 36 evaluation tasks (18 retail, 18 airline).
>
> **Updated models.** While rerunning, we updated to the latest model from each evaluated system — most notably OpenAI's gpt-realtime-1.5 (Feb 2026), a substantial upgrade over the Aug 2025 model.
>
> **Updated results.** Task completion (Clean / Realistic): Google 29%/24% → 31%/26%, OpenAI 33%/19% → 49%/35%, xAI 42%/30% → 51%/38%, GPT-5 text 80% → 85%. The voice-text gap persists (85% text vs 31–51% voice).
>
> **Expanded analysis.** Qualitative error analysis expanded from 40 samples to all 91 failed simulations. Statistical methodology updated to paired permutation tests across retail, airline, and telecom domains (Holm-Bonferroni corrected).
>
> **Updated ablations and voice interaction quality.** OpenAI's new model went from slowest (2.22s) to fastest (0.90s) latency and 68%→100% responsiveness, but selectivity collapsed (74%→6%). Ablation patterns shifted: turn-taking now hurts Google most (-12pp, was \-1pp), while xAI improves \+4pp with turn-taking (was \-6pp). OpenAI became much more robust to noise (-4pp vs \-13pp).
>
> ## W2 \+ Q1: What makes the evaluation specifically full-duplex?
>
> Full-duplex evaluation is central to the paper's contribution. In a half-duplex benchmark, the user sends a message and waits. In τ-Voice, both parties exchange audio simultaneously every tick (Section 3.1), and the evaluation explicitly measures how agents handle the dynamics that arise:
>
> (a) Interruption handling — agent must yield appropriately (yield rate, no-yield errors). (b) Backchannel evaluation — agent must continue through backchannels (backchannel selectivity). (c) Non-directed speech/vocal tics — agent must ignore coughs, asides (selectivity metrics). (d) Simultaneous speech — linearization algorithm for overlapping audio (Appendix B.3). (e) Voice interaction quality metrics (Section 4.5) — responsiveness, latency, interrupt rate, selectivity — meaningless in half-duplex.
>
> The ablation in Table 7 (+Interrupts, renamed to \+Turn-taking in the updated version) directly measures how full-duplex dynamics affect task completion. Table 1 summarizes how these capabilities differ from existing benchmarks — none of which combine all three dimensions.
>
> ## W3: Motivation for integrated evaluation
>
> Task completion and conversational dynamics interact: an agent that interrupts users frequently may miss critical information, causing task failures. Prior work evaluates these in isolation (τ-bench: tasks without voice; Full-Duplex-Bench: turn-taking without tasks). Our integrated evaluation shows turn-taking alone costs 8pp in task completion; the timeline walkthrough (Figure 4\) demonstrates how voice dynamics cascade into task failures.
>
> ## Q2: Task construction
>
> The 278 tasks are inherited from τ2-bench across retail (114), airline (50), and telecom (114), each with gold-standard database state changes. We added voice-specific instructions (spelling behavior, known/unknown info, persona) on top. We will clarify this in the introduction.
>
> ## Q3 \+ Q4: Simulator realism and LLM strength
>
> Motivated by this question, we conducted a human annotation study. Two annotators independently rated 60 simulations on realism (1–4 scale). The simulator achieved an overall mean of 3.1/4, with 83% of ratings at 3 or above. Within-1 inter-rater agreement was 94%. Turn-taking naturalness scored 3.1/4, interruption behavior 3.0/4, and backchannel naturalness 3.5/4. The weakest dimension was voice prosody (2.6/4).
>
> Regarding the strong LLM (GPT-4.1): a weaker LLM would introduce simulator errors that contaminate agent evaluation — even with GPT-4.1, 14 of 91 failures were user simulator errors. The simulator has better memory than real users (Section 6.1), meaning our results are optimistic.
>
> ## W4 \+ W1: Baselines and related work
>
> We surveyed \~10 audio-native models; only 4 support both realtime streaming and native tool calling, all commercial. The rest (Moshi, PersonaPlex, MiniCPM-o 4.5, Qwen-Omni-Realtime) lack native tool calling. At submission we evaluated 3 of 4\. Our fully open-source codebase and adapter architecture (Section 3.1) make adding new models straightforward; we will maintain an updated leaderboard. We also ran a cascaded pipeline (Deepgram STT → GPT-4.1 → ElevenLabs TTS): 63.2% Clean, 43.0% Realistic on Retail — within the audio-native range, confirming the gap holds across architectures. We did not include this because cascaded results reflect one specific pipeline configuration rather than cascaded systems as a class.

---

> > ### Author Rebuttal · Reviewer_macs · 2026-04-04
> >
> > Thank you for your rebuttal. The reviewer provides the following comments:
> >
> > The primary concern lies in the inclusion of full-duplex models. Although the paper aims to evaluate full-duplex systems, several important and representative models are not covered. The reviewer lists a few recent open-sourced works as examples:
> >
> > NTPP: Generative Speech Language Modeling for Dual-Channel Spoken Dialogue via Next-Token-Pair Prediction (ICML 2025)
> > LLaMA-Omni: Seamless Speech Interaction with Large Language Models (ICLR 2025)
> > SALMONN-omni: A Standalone Speech LLM without Codec Injection for Full-duplex Conversation (NeurIPS 2025)
> >
> > The reviewer believes that many other full-duplex speech models with publicly available code and checkpoints exist, yet only a limited subset is included in this paper.
> >
> > Overall, while the reviewer appreciates the authors’ response and is willing to slightly increase the score, they tend to maintain their overall rating and stance, as the contribution of the paper remains limited despite some improvements.

---

> > > ### Author Response · Authors · 2026-04-08
> > >
> > > We thank the reviewer for the specific model suggestions. We investigated all three:
> > >
> > > NTPP (ICML 2025): A generative modeling paradigm for dual-channel spoken dialogue focused on turn-taking and response coherence. Tool calling is not mentioned as supported in the research paper or main documentation.
> > > LLaMA-Omni (ICLR 2025): Built on Llama-3.1-8B-Instruct (which supports tool calling in text mode), but the speech pipeline is trained for speech-to-speech interaction. Tool calling is not mentioned as supported in the research paper or main documentation.
> > > SALMONN-omni (NeurIPS 2025): Built on Llama-3-8B-Instruct with LoRA fine-tuning for full-duplex conversation. The fine-tuning targets conversational alignment and state prediction. Tool calling is not mentioned as supported in the research paper or main documentation.
> > >
> > >
> > > This highlights an important distinction: τ-Voice evaluates full-duplex voice agents, not full-duplex speech models. The difference is tool calling — enabling interaction with external systems (databases, APIs) during the conversation. This is what makes τ-Voice's tasks grounded and verifiable: task success is determined by whether the agent correctly modified database state through tool calls, not by the quality of its speech output alone. We will update the camera ready version to emphasize the importance of tool calling capabilities for inclusion in this benchmark.
> > >
> > > The models the reviewer lists are valuable contributions to full-duplex speech modeling, but they operate in a different evaluation space. They can be assessed on conversational dynamics (turn-taking, backchannels) — which benchmarks like Full-Duplex-Bench address — but they lack the ability to call tools. As future versions of these models add tool-calling support, τ-Voice's adapter architecture (Section 3.1) is designed to incorporate them.
> > >
> > > We surveyed all available audio-native models and their capabilities in our initial rebuttal. Of these, only models with both realtime streaming AND native tool calling can be evaluated on τ-Voice. At submission time, this included four commercial providers; we didn’t add Nova as its another closed source model and we dont expect to learn anything new there.

---

### Official Review · Reviewer_beyN · 2026-03-19

**Soundness:** 3
**Presentation:** 2
**Significance:** 3
**Originality:** 3
**Overall Recommendation:** 4
**Confidence:** 3

**Summary:**

This paper introduces $\tau$-Voice, a benchmark for full-duplex voice agents on grounded customer-service tasks. It extends τ²-Bench with a full-duplex orchestrator, a controllable voice user simulator with accents/noise/turn-taking behaviors, and evaluation of both task completion and voice interaction quality over 278 tasks in retail, airline, and telecom. The main empirical claim is a large gap between text and voice performance, with voice systems reaching only 29–42% pass@1 in clean settings and 19–30% in realistic settings, versus 80% for the GPT-5 text baseline.

**Compliance With Llm Reviewing Policy:**

Affirmed.

**Key Questions For Authors:**

1. Figure 2 is not referred in the paper, and numbers in Table 7 seem to be wrongly computed, please check all Tables again and make the right claim for the manuscript.
2. Why are cascaded baselines such as ASR→LLM→TTS absent from the main comparison, given that the paper explicitly argues future work should include them? Without such baselines, it is difficult to know whether the reported gap is due to the voice modality itself or current end-to-end API design choices. It would be great if authors can provide a simple investigation in the rebuttal section.
3. The simulator receives the agent transcript directly rather than perceiving only the generated speech. Can the authors better quantify how much this simplification changes outcomes?
4. The paper claims novelty from combining three axes. Could the authors explicitly compare $\tau$-Voice against VoiceAgentBench, and articulate what benchmark capability τ-Voice adds that those frameworks do not already provide?
5. Why are repeated runs and confidence intervals emphasized only for Retail?

**Limitations:**

Its realism is partial rather than end-to-end: it is English-only, TTS-based, uses direct transcript injection on the agent-to-user side, and relies on a simulator that is more patient and more capable than real callers, which likely makes the setting easier and cleaner than real deployment. The benchmark is also narrower than the title may suggest, since it remains confined to three inherited customer-service domains and three proprietary audio-native APIs, with no main-result comparison to cascaded or open systems. Some core evaluation choices. especially LLM-based communication verification, limited repeated-run statistics outside Retail, and small-sample qualitative analysis. reduce methodological confidence.

**Strengths And Weaknesses:**

Strengths
1. Benchmark intersection is genuinely novel.
2. The benchmark has 278 inherited grounded task, and the central contribution lies in the simulator, full-duplex orchestration, controllable realism, and dual evaluation axes.
3. The simulator design exposes accents, environmental noise, packet loss, muffling, backchannels, interruptions, and non-agent-directed speech in a controlled way, and the evaluation goes beyond pass@1 to include responsiveness, latency, interruption rate, and selectivity.

Weaknesses
1. The benchmark evaluates English only, uses TTS rather than recorded human speech, feeds the agent transcript directly to the user simulator instead of making the simulator perceive the agent only through speech, and uses a simulator that the paper itself acknowledges is more patient than real users.
2. It covers only three customer-service domains, inherits its task structure from $\tau^{2}$-Bench, and evaluates only three proprietary audio-native providers. There are no open-source or cascaded ASR→LLM→TTS baselines in the main experiments, even though the paper itself notes this as future work. That makes it harder to tell whether the measured gap is really about “voice agents” broadly, or mostly about a very specific set of frontier end-to-end APIs under one simulation regime.
3. The paper says task completion is “fully verifiable,” but for spoken interactions it uses LLM evaluation to verify some agent communications instead of exact matching, which introduces judge dependence.
4. Statistical reliability is also somewhat limited: repeated runs and confidence intervals are reported explicitly only for Retail, not for the full benchmark.
5. Some numbers computed in the Table are wrong and a Figure is placed without being referred.

---

> ### Author Rebuttal · Authors · 2026-03-31
>
> Thank you for your valuable feedback. Since the original submission, the τ2-bench evaluation suite was updated with task fixes, which required us to rerun all experiments. We took this opportunity to also update to the latest models and expand our analysis. We summarize the key changes below before addressing your comments.
>
> ## Summary of key improvements
>
> **Adopted task fixes.** The τ2-bench community (tau2-bench-verified audit, external PRs) identified and fixed 36 evaluation tasks (18 retail, 18 airline).
>
> **Updated models.** While rerunning, we updated to the latest model from each evaluated system — most notably OpenAI's gpt-realtime-1.5 (Feb 2026), a substantial upgrade over the Aug 2025 model.
>
> **Updated results (from both the above changes).** Task completion (Clean / Realistic): Google 29%/24% → 31%/26%, OpenAI 33%/19% → 49%/35%, xAI 42%/30% → 51%/38%, GPT-5 text 80% → 85%. The voice-text gap persists (85% text vs 31–51% voice).
>
> **Expanded analysis.** Qualitative error analysis expanded from 40 samples to all 91 failed simulations. Statistical methodology updated to paired permutation tests across retail, airline, and telecom domains (Holm-Bonferroni corrected).
>
> **Updated ablations and voice interaction quality.** OpenAI's new model went from slowest (2.22s) to fastest (0.90s) latency and 68%→100% responsiveness, but selectivity collapsed (74%→6%). Ablation patterns shifted: turn-taking now hurts Google most (-12pp, was \-1pp), while xAI improves \+4pp with turn-taking (was \-6pp). OpenAI became much more robust to noise (-4pp vs \-13pp).
>
> ## Q1: Table 7 and Figure 2
>
> We apologize — Table 7 had rounding inconsistencies and Figure 2 was not referenced. Both corrected in the updated version.
>
> ## Q2: Cascaded baselines (ASR→LLM→TTS)
>
> Establishing a novel benchmark requires clean, unconfounded baselines of model capabilities. End-to-end models (OpenAI, Gemini, xAI) are opinionated, fully integrated systems where what you evaluate is what gets deployed. Cascaded systems are highly configurable — ASR model, LLM, TTS, VAD, turn-taking policy, silence thresholds — and no standard representation exists, making them a less clean baseline.
>
> That said, our system is fully compatible with cascaded pipelines via its adapter architecture (Section 3.1), and we will make available code to integrate with LiveKit to help the community efficiently explore this space. As a preliminary investigation, we ran a cascaded pipeline (Deepgram Nova-3 STT → GPT-4.1 → ElevenLabs TTS via LiveKit) on Retail (114 tasks): Clean 63.2%, Realistic 43.0% — within the audio-native range (45–71% Clean, 30–45% Realistic), while text GPT-4.1 achieves 76%. We did not include this in the paper because one configuration cannot represent cascaded systems as a class, but it confirms the voice-task gap is not specific to end-to-end APIs.
>
> ## Q3: Transcript injection
>
> We deliberately use the agent's native transcript rather than adding ASR, which would introduce user ASR errors that compound with real agent errors. We chose not to evaluate speech generation quality — prosody, tone, and naturalness are difficult to assess automatically. For future text-free models, our architecture supports adding a cascaded ASR step (Section 6.2).
>
> ## Q4: VoiceAgentBench comparison
>
> VoiceAgentBench (Jain et al., 2025\) plays pre-recorded TTS queries and measures tool invocation accuracy via scripted dialogues. τ-Voice adds full-duplex real-time streaming, verifiable DB state changes from dynamic conversations, realistic audio degradation, turn-taking dynamics, and a voice user simulator. The benchmarks are complementary: VoiceAgentBench tests whether a model can parse a spoken query into the right tool call; τ-Voice tests whether an agent can complete an entire multi-turn voice conversation that modifies real state. We will add this comparison to the camera-ready.
>
> ## Q5: Statistical reliability
>
> Retail was the primary evaluation domain (Section 4.1); time constraints prevented repeated trials across all domains before submission. The updated version extends to all three domains (2 runs per condition, paired permutation tests, Holm-Bonferroni corrected). The key gaps (text-to-voice, clean-to-realistic) are significant across all three domains.
>
> ## W3: LLM evaluation
>
> The majority of evaluation is deterministic DB state comparison — fully verifiable. LLM evaluation is only used for narrow natural language assertions — targeted binary checks used as robust fuzzy string matching, not open-ended judgments. This is a deliberate improvement over the exact string matching in τ-bench, which is a known community pain point; voice output has inherent surface form variability ("four forty-five" vs "$445") that makes exact matching unreliable.
>
> ## Acknowledged limitations
>
> These are discussed in Section 6.1. TTS is more enunciated than real callers (lower bound for difficulty), and the patient simulator means our results are optimistic.

---

> > ### Author Rebuttal · Reviewer_beyN · 2026-04-03
> >
> > I acknowledge the author's reply and the author covers most part of my questions in the rebuttal. However, my main remaining concern is transcript injection: while I understand why the authors bypass agent-side ASR for controllability, this still makes the benchmark somewhat optimistic and means it should be framed more carefully as a controlled benchmark for grounded full-duplex voice interaction rather than a fully end-to-end deployed-agent benchmark. Overall, the rebuttal addresses several concerns well enough that I remain borderline-positive on the paper.

---

> > > ### Author Response · Authors · 2026-04-08
> > >
> > > We appreciate the reviewer's constructive engagement. To directly address the concern that transcript injection makes the benchmark optimistic, we note that the framework supports running evaluation with user-side ASR enabled — having the user simulator perceive the agent's speech through ASR rather than reading the native transcript directly. We ran this evaluation, and as expected, all scores dropped (the tasks become harder and the user simulator becomes less reliable with ASR noise in the loop). Crucially, the statistically significant findings from the non-ASR evaluation — the text-to-voice gap and the clean-to-realistic degradation both hold under the ASR configuration — the benchmark still reliably distinguishes which systems are stronger, even under the more challenging ASR configuration. We will include the full ASR-enabled results in a new appendix section in the camera-ready.
> > >
> > > We agree with the reviewer's suggested framing and will update the paper to make this explicit — specifically in Section 3.1 (orchestrator description) and Section 6.1 (limitations), clarifying that the default configuration evaluates agent reasoning and conversational dynamics under controlled transcript conditions, while the ASR-enabled configuration is available as a stricter end-to-end evaluation mode. We will also highlight explicitly that the default configuration's results represent an upper bound on real-world performance, due to both transcript injection and the use of TTS rather than recorded human speech.

---

### Decision · Program_Chairs · 2026-04-30

**Decision:**

Accept (regular)

**Comment:**

There was some disagreement from reviewers, with one reviewer calling for reject and one calling for accept, two calling for weak accept.
I found the reviewer calling for reject didn't provide solid enough reasoning for the score, where the positive reviewers were more convincing, despite all reviewers finding some methodological weaknesses in the approach.

Accordingly, weak accept seems the right score.